# Control Reinforcement Learning: Interpretable Token-Level Steering of LLMs via Sparse Autoencoder Features

## Abstract

Large language models exhibit emergent misalignment behaviors during test-time generation, necessitating dynamic control mechanisms for safe deployment. Inspired by sparse interpretable representations, sparse autoencoders (SAEs) can disentangle monosemantic features from superpositioned dense activations, offering a natural interface for controlling language model behavior through interpretable feature manipulation. This work introduces Control Reinforcement Learning (CRL), a framework to unify reinforcement learning with SAE features for interpretable token-level language model control. CRL enables interpretable branch tracking by isolating feature contributions at each generation step, revealing which features drive behavior changes across diverse benchmarks including question answering, bias mitigation, and reasoning tasks. To balance exploration and exploitation, the framework employs Adaptive Feature Masking (AFM) to encourage diverse yet effective feature discovery while maintaining interpretability. Through token-wise feature analysis, CRL provides mechanistic insights into model behavior, revealing task-specific feature contributions across diverse benchmarks including question answering, bias mitigation, and safety tasks. The framework is compatible with supervised fine-tuning, providing complementary control when applied to SFT models. Results demonstrate that interpretable steering can serve as both a control method and analysis tool, establishing a practical pathway for controllable AI systems. [1]

## 1 Introduction

Large language models exhibit emergent misalignment behaviors during test-time generation, where models may deviate from intended objectives despite extensive training (Betley et al., 2025; Casademunt et al., 2025). Fine-tuning approaches often introduce unintended side effects or unexpected behavioral changes (Qi et al., 2024), while existing reinforcement learning methods fail to account for per-token contributions, thus limiting interpretability. Recent work in mechanistic interpretability has shown that sparse autoencoders (SAEs) can extract monosemantic features from neural activations (Bricken et al., 2023), decomposing each token's residual stream into an interpretable feature dictionary. SAE-extracted features enable interpretable steering without modifying base parameters (Durmus et al., 2024). However, static steering approaches cannot address the dynamic nature of misalignment that emerges token-by-token during generation, necessitating adaptive control mechanisms.

However, existing SAE-based steering approaches face significant limitations: (1) contrastive datasets (Soo et al., 2025) or large activation storage (Zhao et al., 2025) are required to identify steering directions, and (2) they rely on context tokens to select features and coefficients. Consequently, current applications have been restricted to specific domains such as bias mitigation and jailbreaking prevention (Durmus et al., 2024; O'Brien et al., 2025). Moreover, SAE feature selection in these applications does not directly capture language models' generation capabilities.

To address these limitations, this work introduces **Control Reinforcement Learning (CRL)**, a dynamic steering framework that applies reinforcement learning-based control for token-wise feature

---

[1] Code, trained models, and all interactive steering demos will be released upon acceptance.

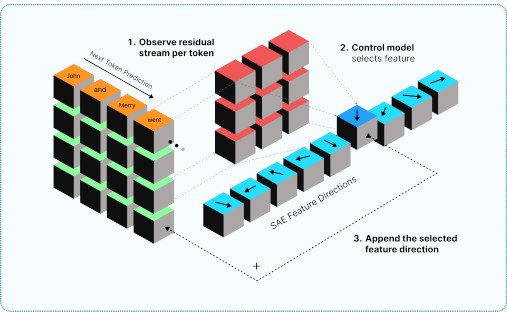 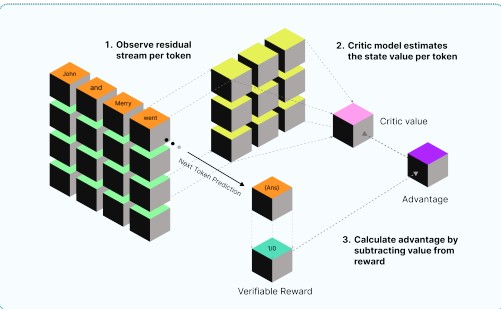

Figure 1: Overview of the Control Reinforcement Learning (CRL) framework. The policy network (left) selects interpretable SAE features at each token to steer generation, while the critic network (right) estimates state values to guide optimization. The framework is formulated as an MDP that observes only the current residual stream state.

selection. This enables token-level control by applying interpretable feedback actions at each generation step, learning to rapidly identify task-relevant features that drive performance. Our contributions are: (1) extending interpretable steering to general benchmarks, improving accuracy without updating base model weights, (2) developing CRL for token-wise control, and (3) introducing Adaptive Feature Masking (AFM) for diverse feature discovery.

## 2 RELATED WORK

**Sparse Autoencoders** address the superposition hypothesis (Elhage et al., 2022) by learning to decompose neural activations into interpretable, sparse features (Huben et al., 2023; Bricken et al., 2023). Given an activation vector $\mathbf{x} \in \mathbb{R}^d$, an SAE learns an encoder $f_{\text{enc}} : \mathbb{R}^d \to \mathbb{R}^{d_{dict}}$ and decoder $f_{\text{dec}} : \mathbb{R}^{d_{dict}} \to \mathbb{R}^d$ where $d_{dict} \gg d$, such that:

$$\mathbf{z} = f_{\text{enc}}(\mathbf{x}) = \text{Activation}(\mathbf{W}_{\text{enc}}\mathbf{x} + \mathbf{b}_{\text{enc}}) \tag{1}$$

$$\hat{\mathbf{x}} = f_{\text{dec}}(\mathbf{z}) = \mathbf{W}_{\text{dec}}\mathbf{z} + \mathbf{b}_{\text{dec}} \tag{2}$$

The training objective uses reconstruction loss and sparsity regularization: $\mathcal{L} = \|\mathbf{x} - \hat{\mathbf{x}}\|^2 + \lambda\|\mathbf{z}\|_1$.

**Activation Engineering** techniques work by making targeted perturbations to a model's activations (Rawte et al., 2023; Turner et al., 2023; Hernandez et al., 2024; Rimsky et al., 2024). In Activation Engineering, steering vectors (Subramani et al., 2022; Konen et al., 2024) enable control over the language model's behavior, offering more direct behavioral control compared to prompt engineering. In this context, SAE features not only enhance interpretability but can also serve as steering vectors in clamp or addition operations (Chalnev et al., 2024). Selecting a suitable coefficient for the generated steering vector is crucial for keeping the language model within its optimal "sweet spot" without disruption (Durmus et al., 2024). Typically, quantile-based adjustments or handcrafted coefficients are common methods for regulating a feature's coefficient (Choi et al., 2024).

**SAE-based Control** methods demonstrate that SAE features can serve as steering mechanisms for language model control. However, current approaches face significant limitations: they require either contrastive datasets for feature selection or extensive activation storage for coefficient optimization. More critically, existing methods lack adaptive feedback mechanisms that can adjust steering strategies based on generation quality, limiting their effectiveness across diverse tasks.

**Reinforcement Learning for LLM** has been explored in various contexts (Yu et al., 2017; DeepSeek-AI, 2025), primarily focusing on high-level policy optimization. Recent work has applied RL to interpretable model steering (Ferrao, 2024), but with limited action spaces. LLMs operate with dual sequences: token positions (Text Stream) and layer information flow (Residual Stream), both conceptualizable as Markov processes. This suggests potential for RL-based SAE control that automatically identifies optimal feature manipulations to maximize task-specific rewards.

## 3 METHOD

### 3.1 PROBLEM FORMULATION: CRL AS AN MDP OVER SAE FEATURES

Figure 1 provides an overview of our Control Reinforcement Learning framework, illustrating the integration of policy and critic networks with SAE feature steering. We approximate the control of transformer representations as a Markov Decision Process (MDP) in which sparse autoencoder (SAE) features are manipulated to optimize task-specific rewards. The underlying problem depends on the full history of residual stream activations and is naturally formulated as a Partially Observable MDP (POMDP) (Zhong & Zhang, 2023; Lim et al., 2023). We adopt the MDP formulation, treating the current residual stream activation as the state for reinforcement learning (Yu et al., 2017; Gui et al., 2019; Xu & Jin, 2024).

Let $\mathbf{x} \in \mathbb{R}^d$ denote the residual stream activations at layer $\ell$ for a target token position, where $d$ is the hidden dimension of the transformer model. Given a pre-trained SAE with encoder $\mathbf{W}_{enc} \in \mathbb{R}^{d \times d_{dict}}$ and decoder $\mathbf{W}_{dec} \in \mathbb{R}^{d_{dict} \times d}$, the sparse feature activations are computed as $\mathbf{z}_t = \text{Act}(\mathbf{x}_t \mathbf{W}_{enc} + \mathbf{b}_{enc})$, where $\text{Act}(\cdot)$ denotes a generic pointwise activation used in the SAE encoder and $\mathbf{z} \in \mathbb{R}^{d_{dict}}$ represents the sparse feature activations with dictionary size $d_{dict}$.

The MDP is defined by the tuple $(\mathcal{S}, \mathcal{A}, \mathcal{P}, \mathcal{R})$ where:

- **State Space** $\mathcal{S}$: The observation is $\mathbf{s} = \mathbf{x} \in \mathbb{R}^d$, the residual stream activation at layer $\ell$ for the current token position.
- **Action Space** $\mathcal{A}$: Actions are binary vectors $\mathbf{a} \in \{0, 1\}^{d_{dict}}$ where selected features (via argmax or top-k) are set to 1, reducing the exploration challenge in high-dimensional feature spaces.
- **Transition Function** $\mathcal{P}$: Deterministic transition governed by the transformer's forward pass with steering applied.
- **Reward Function** $\mathcal{R}$: Task-specific rewards $r$ based on final output quality evaluation.

Although the full problem is a POMDP, we empirically find that treating the current residual stream as a sufficient statistic is effective, making the MDP formulation a practical approximation.

### 3.2 CRL-TOKEN: TOKEN-LEVEL INTERPRETABLE STEERING VIA SAE FEATURES

At each generation step $t$, CRL-Token performs token-level interventions by observing the current token's residual stream activation at layer $\ell$ and applying perturbations:

$$\tilde{\mathbf{x}}_t = \mathbf{x}_t + c \cdot \mathbf{a}_t \mathbf{W}_{dec}, \tag{3}$$

where $\mathbf{a}_t \in \{0, 1\}^{d_{dict}}$ is a sparse binary selection vector, $c$ is the steering coefficient, and $\tilde{\mathbf{x}}_t$ is the steered activation. By the Markov property, the policy depends only on the current residual stream activation:

$$\pi_\theta(\mathbf{a}_t \mid \mathbf{x}_{1:t}) = \pi_\theta(\mathbf{a}_t \mid \mathbf{x}_t),$$

where $t$ denotes the token generation step, starting from the first generated token after the input context.

#### 3.2.1 POLICY NETWORK FOR FEATURE SELECTION

The policy network $\pi_\theta : \mathbb{R}^d \to \mathbb{R}^{d_{dict}}$ maps residual stream observations to SAE feature selection logits. We implement this as an MLP:

$$\boldsymbol{\mu} = \pi_\theta(\mathbf{s}) \tag{4}$$
$$\mathbf{a} = \text{TopK}(\boldsymbol{\mu}, k) \tag{5}$$

The policy first computes feature selection logits, then selects the top-k features. In this study, we set $k = 1$ to focus on the most relevant feature per token, effectively equivalent to ArgMax selection. This design prioritizes interpretability, enabling clear attribution of steering effects to individual features. We employ a softmax parameterization for the policy, converting logits $\boldsymbol{\mu} = \pi_\theta(\mathbf{s})$ into probabilities $p_j = \frac{\exp(\mu_j)}{\sum_{i=1}^{d_{dict}} \exp(\mu_i)}$. This parameterization ensures differentiability and enables PPO (Schulman et al., 2017) to optimize the expected advantage using the log-probability of the selected feature:

$\log \pi_\theta(\mathbf{a}|\mathbf{s}) = \log p_{j^\star}$, where $j^\star = \arg\max_j \mu_j$. By decoupling probability learning from coefficient determination, the model can learn which features to select without prematurely collapsing to a single feature.

**Partial Observability:** CRL-Token is formulated as a POMDP because the policy observes only the residual stream at a specific layer without access to the sampled token information, yet the next state depends on the token that will be sampled and added to the KV cache. This creates partial observability where undetermined token states affect future transitions. While empirical injectivity of layer representations (Nikolaou et al., 2025) suggests that hidden states at arbitrary layers contain sufficient information about input tokens, we use temperature=0 sampling to eliminate stochastic token sampling effects during training and evaluation.

### 3.3 Adaptive Feature Masking

CRL-Token introduces Adaptive Feature Masking (AFM), a mechanism designed to balance exploration and exploitation during multi-token generation. AFM maintains a per-sample boolean mask over the SAE dictionary, where $\text{mask}_i = 1$ indicates feature $i$ is available for selection. At the start of each sample, the mask is initialized to a small subset (e.g., frequently active features), and progressively expanded during generation:

$$\text{mask}_i^{(t+1)} = \text{mask}_i^{(t)} \vee \text{active}_i^{(t)}, \quad \mathbf{a}_t = \text{TopK}(\boldsymbol{\mu} \odot \text{mask}^{(t)}, k) \tag{6}$$

where $\text{active}_i^{(t)} = [\mathbf{z}_{t,i} > 0]$ indicates whether feature $i$ was activated at step $t$, and the mask constrains policy selection by masking unavailable features. This per-sample masking prevents the policy from collapsing to features that were frequently selected during early training, encouraging each sample to explore task-relevant features dynamically while allowing the policy to build on previously activated features within the same generation trajectory.

### 3.4 Critic Network for Value Estimation

The critic $V_\phi : \mathbb{R}^d \to \mathbb{R}$ estimates the state value function:

$$V_\phi(\mathbf{s}) = \mathbb{E}_{\pi_\theta}[r \mid \mathbf{s}], \tag{7}$$

and is implemented as a multilayer perceptron (MLP).

### 3.5 Optimization with PPO and Task-Specific Rewards

The steering coefficient $c$ is determined adaptively by averaging activation magnitudes from correctly predicted training samples, eliminating the need for manual tuning across different layers and tasks.

We train both the policy and critic using Proximal Policy Optimization (PPO). For feature selection via argmax/top-$k$, the PPO objectives are:

$$L_{\text{policy}}(\theta) = \mathbb{E}\left[\min\left(r_t(\theta)A_t, \text{clip}(r_t(\theta), 1-\epsilon, 1+\epsilon)A_t\right)\right], \tag{8}$$

$$L_{\text{critic}}(\phi) = \mathbb{E}\left[(V_\phi(\mathbf{s}) - r)^2\right]. \tag{9}$$

Here, $\pi_\theta(\mathbf{a}|\mathbf{s})$ is computed from the softmax probabilities described above, $r_t(\theta) = \frac{\pi_\theta(\mathbf{a}|\mathbf{s})}{\pi_{\theta_{\text{old}}}(\mathbf{a}|\mathbf{s})}$ is the probability ratio for the selected feature, and $A_t = r - V_\phi(\mathbf{s})$ is the advantage estimate.

The reward $r$ is task-specific. For multiple-choice tasks, we use binary rewards following DeepSeek-Math (Shao et al., 2024): $r(\hat{y}, y^*) = 1$ if $\hat{y} = y^*$, and 0 otherwise. For reasoning and safety tasks, we use task-specific correctness and refusal metrics, with details provided in Appendix A.

### 3.6 Experimental Setup

We evaluate CRL using Gemma-2 2B-IT (Team, 2024a) with Gemma Scope SAEs (Lieberum et al., 2024) across diverse benchmarks including knowledge (MMLU (Hendrycks et al., 2021)), reasoning (GSM8K (Cobbe et al., 2021)), bias (BBQ (Parrish et al., 2022)), and safety (HarmBench (Mazeika et al., 2024), XSTest (Röttger et al., 2024)) tasks. The framework is trained using PPO with task-specific rewards to optimize feature selection for improved performance while maintaining interpretability. Complete experimental details are provided in Appendix A.

## 4 RESULTS

### 4.1 TASK-SPECIFIC PERFORMANCE GAINS

Our Control Reinforcement Learning (CRL) approach demonstrates consistent but modest performance improvements across benchmarks. The results show gains over baseline models, with improvements in safety and bias mitigation tasks. We report full numbers and layer/coefficent sweeps to make effects auditable. CRL-Token adaptively selects SAE features for steering based on each token's residual stream activation, enabling dynamic token-level control during generation. This approach provides inherent interpretability by revealing each token's dedicated features, which we analyze in detail in Section 5.3. Table 1 reports the main results for the Gemma 2 2B model using the standard **single-layer** CRL-Token configuration, where one intervention layer is selected for steering.

Table 1: Performance results for Gemma 2 2B model across different tasks using **single-layer** CRL-Token. The table shows task type, intervention layer, baseline accuracy (Before), CRL accuracy (After), and gain in percentage points. Layer column shows where task-relevant features emerge for diagnostic analysis. Results reported as mean $\pm$ std across three seeds.

| Task | Type | Layer | Before | After | Gain |
|---|---|---|---|---|---|
| MMLU | Multi-choice QA | 24 | $52.06_{\pm0.23}$ | $55.37_{\pm0.16}$ | +3.31 |
| MMLU-Pro | Multi-choice QA | 25 | $30.30_{\pm0.00}$ | $30.49_{\pm0.07}$ | +0.19 |
| BBQ Ambig | Bias QA (ambiguous) | 5 | $60.17_{\pm0.01}$ | $65.86_{\pm3.03}$ | +5.69 |
| BBQ Disambig | Bias QA (disambiguated) | 5 | $84.38_{\pm0.52}$ | $84.85_{\pm0.09}$ | +0.47 |
| SimpleQA | Short-form QA | 8 | $3.78_{\pm0.17}$ | $4.00_{\pm0.29}$ | +0.22 |
| GSM8k | Math reasoning | 24 | $54.62_{\pm0.16}$ | $55.65_{\pm0.33}$ | +1.03 |
| HarmBench | Adversarial safety | 21 | $41.46_{\pm9.05}$ | $49.12_{\pm1.59}$ | +7.66 |
| XSTest | Over-refusal | 12 | $86.35_{\pm0.00}$ | $87.62_{\pm0.82}$ | +1.27 |

*Multi-layer variant.* For completeness, we also experimented with **CRL-Layer**, a variant that coordinates steering across multiple layers using a shared policy network. Under this setting, HarmBench and XSTest scores reach 87.14% and 89.84%, respectively. However, such multi-layer manipulation trades single-feature attribution for stronger aggregate refusal behavior; we therefore present it as an ablation (Appendix A.6) and keep CRL-Token as the interpretable mainline. Therefore, it is reported as an additional analysis in Appendix A.6 rather than as the main configuration.

### 4.2 SENSITIVITY TO STEERING LAYER AND COEFFICIENTS

To isolate the effects of coefficient changes, we conducted systematic coefficient analysis across different steering magnitudes. Based on experimental results shown in Figure 2 for MMLU tasks (extended analysis in Appendix 16), we discovered that optimal coefficients vary significantly with network depth, consistent with our analysis summarized in two key findings:

- **Layer-specific effectiveness**: Certain layers demonstrate higher utility, with optimal intervention points varying across task conditions.
- **Coefficient scaling**: Large coefficients perform poorly in earlier layers, suggesting that aggressive steering during early processing stages disrupts fundamental model capabilities.

Across coefficients ranging from 10 to 100 for CRL-Token steering (Figure 2), later layers outperform earlier layers. Interestingly, large coefficients in early layers yield poor performance, while this phenomenon becomes mixed in later layers. This aligns with prior observations that residual stream norms increase across layers (Heimersheim & Turner, 2023), meaning identical coefficient values produce different effects at each layer. The observed performance inversions indicate that each layer differs in the optimal coefficient range called "sweet spot" (Durmus et al., 2024), necessitating the dynamic approach we employ through averaging observed coefficients. Layer-wise analysis across all coefficient manipulations also reveals an additional pattern. Independent of coefficient values, the overall trend follows layer-specific patterns, indicating that performance depends on layer-specific features. Beyond performance differences, the varying optimal layers indicate that different tasks require distinct feature activations at different network depths, suggesting task-specific intervention strategies. Residual stream norm analysis is provided in Appendix 15.

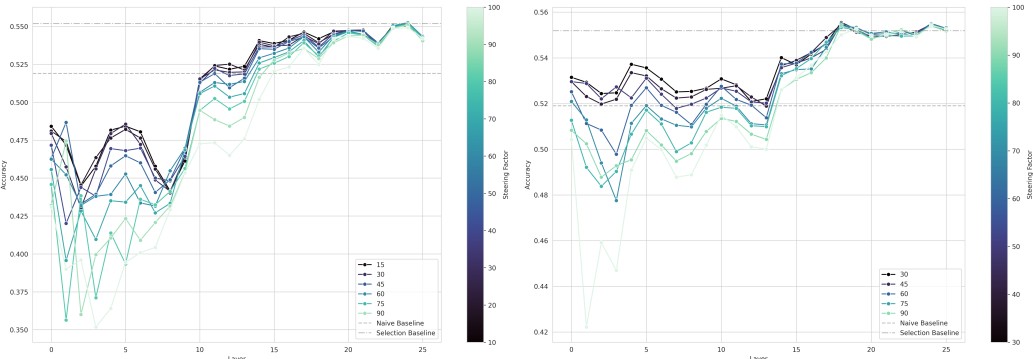

Figure 2: MMLU performance across different layers for Gemma 2 2B model, showing optimal intervention points for unconstrained (left) and constrained (right) decoding (definitions in Appendix A.1).

Further analysis of MMLU layer patterns (Figure 16) illustrates the difference between unconstrained and constrained decoding patterns (definitions in Appendix). The varying optimal layers indicate that hallucination mitigation and factual question answering require distinct feature activations. This pattern demonstrates that identical coefficient values produce different effects at each layer due to residual stream norm variations. BBQ benchmark results from coefficient experiments exhibit similar tendencies to MMLU. Larger coefficients perform poorly in earlier layers while showing mixed performance in later layers. As shown in Figure 17 (in Appendix A), BBQ ambiguous and disambiguated tasks display distinct layer patterns for optimal performance, implying that different bias contexts require distinct feature activations for mitigation. The analysis for layers $\ell \geq 18$ for BBQ tasks (Figure 18) confirms that ambiguous and disambiguated contexts require different optimal intervention layers. This suggests that bias mitigation strategies must be adapted to the specific type of ambiguity present in the input.

The layer-wise analysis reveals a consistent pattern across tasks: later layers generally provide more effective intervention points, while early layers show degraded performance with large coefficients. This pattern holds across different coefficient values and task types, consistent with observations that residual stream norms increase across layers (Heimersheim & Turner, 2023). The consistent emergence of task-specific optimal layers across MMLU, BBQ, and other benchmarks suggests that effective steering requires careful consideration of both the intervention layer and the task characteristics. Extended layer-wise coefficient analysis across all tasks is provided in Appendix A.

### 4.3 CRITIC NETWORK ANALYSIS

The critic network provides valuable insights into CRL's learning dynamics across different task types. We analyze critic behavior patterns for both single-token and multi-token generation tasks.

#### 4.3.1 SINGLE-TOKEN TASKS

For single-token tasks (MMLU, BBQ), critic values show clear distributions between correct/incorrect samples. MMLU exhibits critic bottlenecks where corrected and misguided samples (defined in Appendix) remain nearly indistinguishable, while BBQ demonstrates effective policy-critic coordination with clear sample distinction. Detailed single-token critic analysis is provided in Section 5.1.

#### 4.3.2 MULTI-TOKEN TASKS

For multi-token tasks like GSM8K, the critic network's value function exhibits climbing patterns along token sequences, indicating effective learning of policy feature selection through AFM (Section 3.3). As illustrated in Figure 4, critic value patterns are categorized into four distinct groups. Correct cases display higher gradients with divergence beginning around 200 tokens, while steering-induced cases show different trends with corrected answers exhibiting value increases at the 400-token point. This indicates that CRL-Token's interventions become more effective in later generation stages. Additional multi-token critic analysis including HarmBench and XSTest patterns is provided in Appendix A.8.

## 5 DISCUSSION

### 5.1 CRITIC BOTTLENECKS IN SINGLE-TOKEN DECISIONS

The critic networks exhibit distinctive learning dynamics across different tasks, revealing task-dependent bottlenecks in either policy mechanisms or critic functionality. For single-token tasks (MMLU, BBQ), critic values show clear distributions between correct/incorrect samples, while multi-token tasks (HarmBench, XSTest) exhibit different patterns reflecting reward estimation complexity. As shown in Figure 3, the critic value distributions illustrate these patterns, with distinct clustering between correct and incorrect sample categories. For MMLU, corrected and misguided samples remain nearly indistinguishable, indicating a critic bottleneck. Conversely, for BBQ, corrected and misguided samples become clearly distinguishable, with corrected samples achieving higher values, suggesting effective policy-critic coordination.

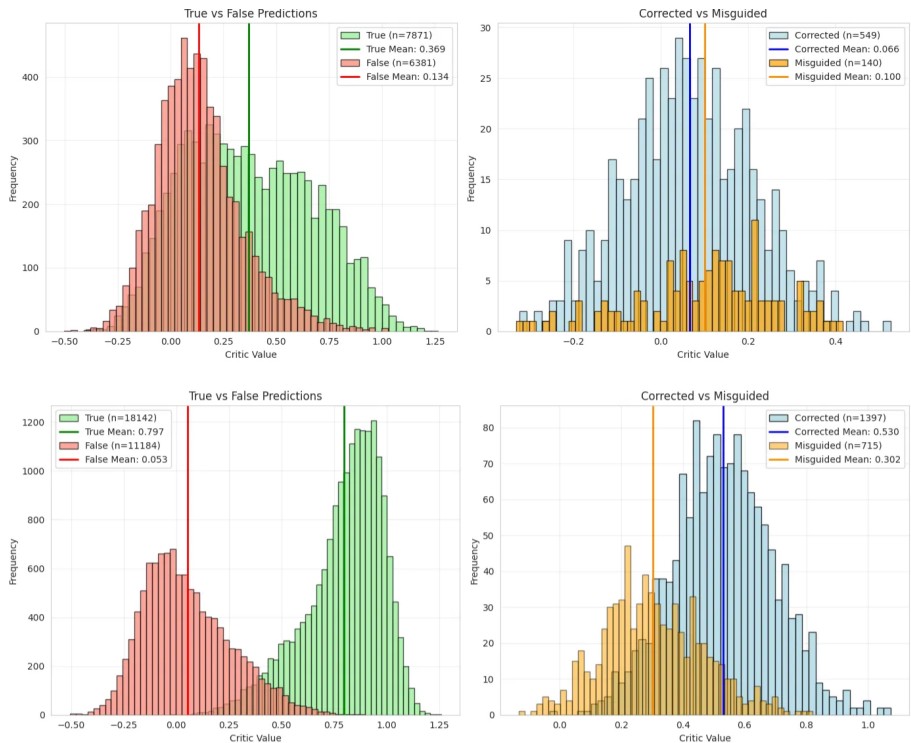

Figure 3: Critic value distributions for single-token tasks. Top: MMLU, Bottom: BBQ. Colors: green = unchanged-correct, red = unchanged-incorrect, blue = corrected, yellow = misguided (see Appendix A.1 for definitions).

### 5.2 TRAJECTORY-BASED CRITIC DIAGNOSTICS FOR MULTI-TOKEN TASKS

The critic value distribution extends analysis to examine critic trends throughout token generation, with critic values predicted from each token's residual stream. As shown in the GSM8K critic analysis (Figure 4), multi-token tasks exhibit distinctive trajectory patterns throughout generation. For HarmBench, the critic successfully distinguishes between correct and incorrect samples, whereas XSTest exhibits inverted estimation patterns. However, while HarmBench shows clear gaps at sequence boundaries with minimal gradient differences, XSTest demonstrates estimation errors in gap measurements but maintains clear superiority for correct samples in slope analysis.

This discrepancy stems from XSTest's task characteristics, which require distinguishing benign requests with higher context dependency, making reward estimation more challenging. Similar to the MMLU case, this indicates a critic bottleneck, while HarmBench achieves accurate estimation but shows limited critic improvement through policy feature selection, suggesting a policy bottleneck. Detailed analysis of HarmBench and XSTest critic patterns is provided in Appendix A.8.

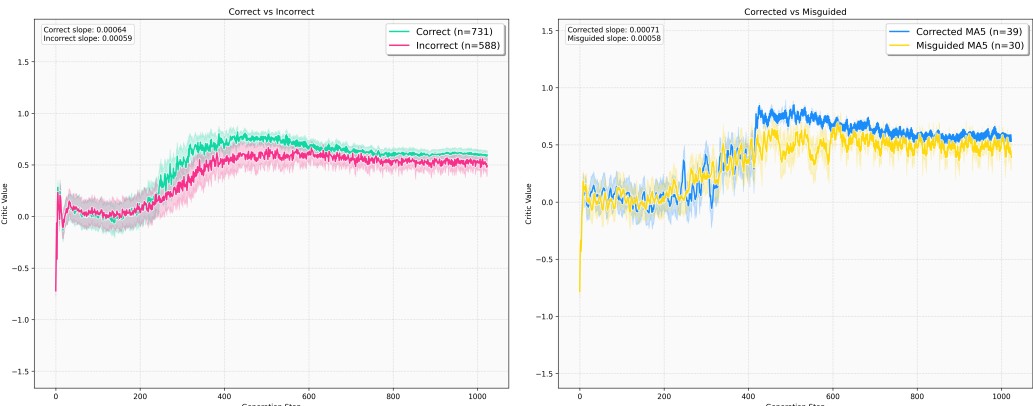

Figure 4: Critic network value trajectories for GSM8k task. Colors indicate: green (unchanged correct), red (unchanged incorrect), blue (corrected), yellow (misguided) (see Appendix A.1).

### 5.3 FEATURE STEERING AS EVIDENCE OF SEMANTIC COHERENCE

Analysis of learned feature selections reveals semantically meaningful patterns aligned with task requirements. The policy network identifies features corresponding to relevant cognitive processes: reasoning, fact retrieval, and safety considerations.

Efficacious steering demonstrates semantic coherence across corrected examples. Figure 5 shows Feature 10961, described as *terms related to statistical methods and implementation details*, consistently activating on "number" and "in" tokens across diverse mathematical contexts. Figure 6 demonstrates Feature 2317, described as *phrases indicating comparisons or relationships between entities*, activating on equality tokens, showing semantic coherence despite varying problem structures. Additional examples illustrating lexical generalization are provided in Appendix B.2.

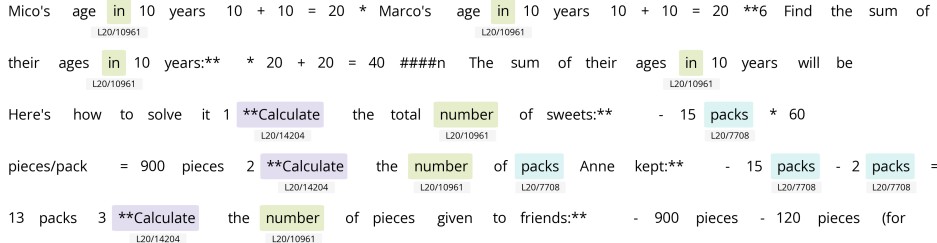

Figure 5: GSM8K corrected case 1: Contextually appropriate feature steering activates relevant numerical tokens, strengthening reasoning.

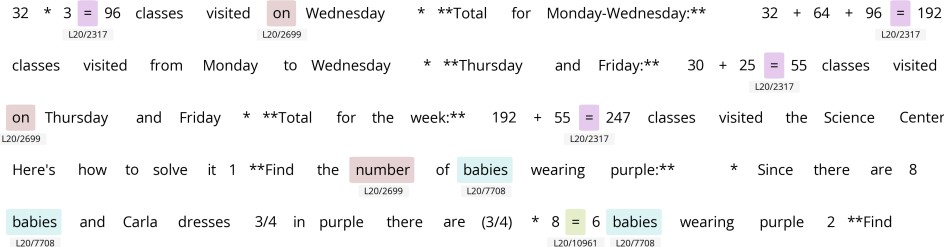

Figure 6: GSM8K corrected case 2: Semantically coherent feature activations align with equality tokens, guiding correct computation.

However, failure cases arise from two primary mechanisms. Figure 7 shows Feature 15434, described as *complex relationships involving socio-legal and psychological themes*, activating on "girls'" token, demonstrating how semantically reasonable features prove unsuitable for mathematical reasoning tasks. Figure 8 reveals token-feature misalignment where features like 1222, described as *sentences expressing emotional vulnerability and complex interpersonal dynamics*, and 4069, described as

*mathematical notation and geometric properties related to circles and angles*, show poor semantic alignment with activated tokens "the" and "score". Figure 20 demonstrates feature interference causing the model to abandon correct solution paths through inappropriate feature activation.

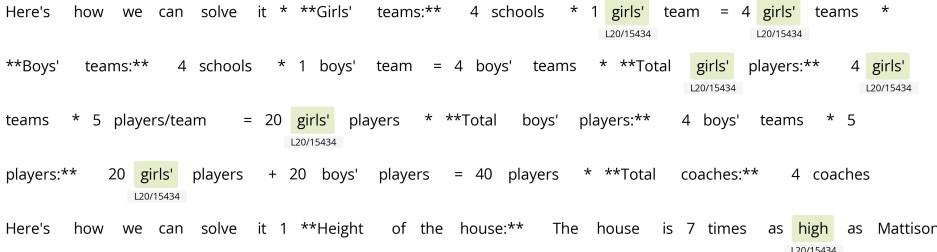

Figure 7: GSM8K misguided case 1: Selection of task-irrelevant features interferes with mathematical reasoning and diverts the solution process.

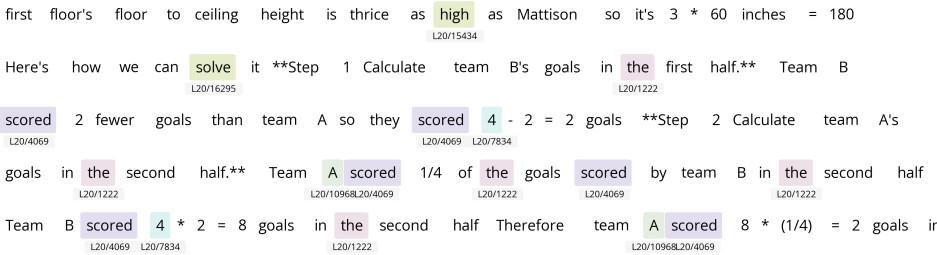

Figure 8: GSM8K misguided case 2: Token-feature misalignment produces semantically incoherent activations, degrading reasoning performance.

### 5.4 BRANCH TRACKING ANALYSIS

Branch point analysis compares cases where the same problem context produces correct or incorrect outputs depending on which SAE features are selected for steering. We trained separate policies on GSM8K at layers 10 and 20 to analyze how steering effectiveness varies across layers. As shown in Figure 9, layer 10 features exhibit concrete, syntactic properties while layer 20 features demonstrate abstract, contextual semantics. Balloon pricing demonstrates this: "1. Find the original price..." activates layer 20's "mathematical proofs" feature recognizing logical derivation structure, whereas "Step 1: Find..." activates layer 10's "summaries" feature treating it as tutorial explanation. Rice consumption with "kg/member" notation activates layer 10's "clinical examinations" feature due to surface similarity with medical dosage, while layer 20's "assessments and strategic plans" captures the planning semantics. Financial terminology like "Initial Share" triggers layer 10's "legal contracts" feature, while "Diamond" and "Gold" in premium calculations activate "faith and personal beliefs" features. Restaurant problems show layer 10's "mathematical notation" feature outperforming layer 20's "food and dining" by prioritizing numerical structure. These patterns suggest matching feature abstraction levels to task requirements. Detailed analysis is provided in Appendix B.5.

### 5.5 FEATURE BEHAVIOR AND FAILURE ANALYSIS

Our experiments reveal insights into SAE feature controllability. While corrected cases demonstrate clear semantic coherence between feature activations and their documented descriptions (as shown in GSM8K examples), misguided cases reveal limitations where features exhibit behavior unrelated to their semantic interpretations. This discrepancy primarily arises in failure modes, where gaps between feature activation patterns and downstream steering effects become apparent. The observed divergence in misguided cases suggests that while SAE features maintain interpretability for successful interventions, failure modes may involve more complex feature interactions. Co-activating features can produce steering behaviors that differ from their individual documented descriptions. Our results suggest that feature interactions may be more complex than simple linear combinations, as isolated feature activation can produce unexpected steering effects. The limited effectiveness in steering complex reasoning tasks indicates that such cognitive functions may require coordinated interventions across multiple components.

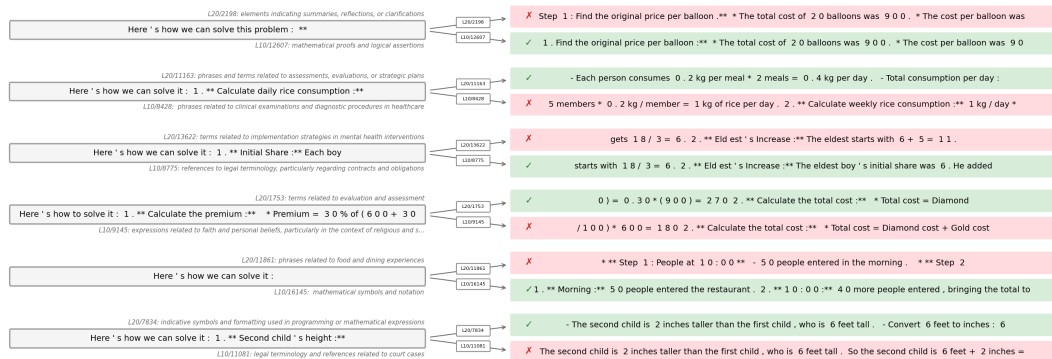

Figure 9: Branch point analysis comparing layer 10 vs layer 20 feature semantics. Each panel shows context and competing features from both layers, with correct features highlighted.

**Bottleneck Analysis.** Our analysis reveals two primary bottlenecks limiting CRL performance: critic network precision and policy's critic value utility. For tasks like MMLU and XSTest, critic bottlenecks manifest as estimation errors where corrected/misguided samples remain nearly indistinguishable, while HarmBench shows policy bottlenecks with accurate critic estimation but limited improvement through feature selection. Conversely, BBQ demonstrates effective policy-critic coordination with clear distinction between sample categories. This suggests that while MMLU policy primarily addresses hallucination issues, BBQ policy effectively tackles bias-related cases within ambiguous scenarios. This task-dependent bottleneck pattern suggests different optimization strategies for different task types. Detailed critic analysis for HarmBench and XSTest is provided in Appendix A.8.

**Failure Mode Analysis.** Failure cases arise from two primary mechanisms: (1) semantically reasonable features prove unsuitable for specific task requirements, and (2) token-feature relationships lack semantic alignment. These observations suggest potential improvements through conditional steering approaches that suppress interventions when feature-token alignments are semantically inappropriate.

**Practical Implications.** CRL demonstrates the potential for model adaptation through task-specific rewards in controlled settings. The framework is compatible with supervised fine-tuning, yielding complementary improvements (e.g., MMLU: 55.23 base → 55.73 SFT → 56.11 SFT+CRL). The observed feature interactions highlight the need for careful evaluation when deploying such steering methods. Detailed feature analysis and safety considerations are provided in Appendix B.

## 6 CONCLUSION AND LIMITATIONS

We present Control Reinforcement Learning (CRL), a framework that trains policy networks for interpretable token-wise feature steering based on task rewards. CRL improves performance across diverse benchmarks—including question answering, bias mitigation, safety, and reasoning—while providing interpretable control through sparse autoencoder (SAE) feature manipulation. The framework is compatible with supervised fine-tuning, providing complementary improvements when applied to SFT models. Analysis of critic value trajectories and feature selections shows how CRL identifies task-relevant directions and reveals task-dependent bottlenecks, highlighting its role as both a control method and an interpretability tool.

Our results also expose challenges in SAE-based steering. Non-monosemantic features can cause unintended behavioral changes, limiting reliable control. Two main failure modes emerge: semantically reasonable features that do not meet task requirements, and token-feature relationships with weak semantic alignment. These findings suggest that feature interactions may exhibit non-additive effects beyond simple linear combinations.

Overall, CRL establishes a practical pathway for interpretable representation-level control while revealing fundamental challenges in mechanistic interpretability and alignment. Future work should develop more monosemantic decomposition methods, improve critic-policy coordination across tasks, and design monitoring tools to detect and prevent unintended effects.

## ETHICS STATEMENT

This work introduces Control Reinforcement Learning (CRL), a framework for interpretable token-level steering of large language models (LLMs) through Sparse Autoencoder (SAE) features. We have carefully considered the broader impacts of this research in line with the ICLR Code of Ethics.

**Contribute to society and human well-being.** CRL is designed to provide interpretable control over model behavior, improving reasoning performance and supporting safer and fairer deployment of LLMs. We explicitly evaluate on safety and bias benchmarks to demonstrate potential societal benefits in alignment and robustness.

**Uphold high standards of scientific excellence.** All benchmarks, datasets, and methods used in this study will be publicly available upon acceptance. We provide algorithmic details, evaluation procedures, and ablations to ensure transparency. No human subjects or private data are used.

**Avoid harm.** While CRL improves refusal behavior and mitigates biases in some cases, we identify failure modes such as misguided feature selection and semantic misalignment that can degrade performance. We also acknowledge the dual-use potential of steering methods: although CRL aims to enhance model alignment, feature manipulation could be misused to induce harmful behaviors.

**Be fair and take action to avoid discrimination.** We evaluate CRL on social bias benchmarks (e.g., BBQ) and show reductions in measured bias. Nonetheless, residual biases inherited from pretrained models remain, highlighting the need for continued auditing and fairness assessment.

**Respect privacy and confidentiality.** This work does not involve human participants, personal data, or confidential information. All datasets are used under their respective licenses.

Overall, CRL advances the development of interpretable and responsible control methods for LLMs. However, we emphasize the importance of further safeguards, auditing, and ethical deployment practices to mitigate risks.

## REPRODUCIBILITY STATEMENT

We have taken extensive steps to ensure reproducibility of our results. All benchmarks used in this study (MMLU, MMLU-PRO, BBQ, GSM8K, HARMBENCH, XSTEST, SIMPLEQA) are publicly available, with dataset splits and evaluation metrics detailed in Appendix A.3. Algorithmic details of CRL, including the Markov Decision Process formulation, PPO training procedure, and reward signal design, are described in Section 3, with pseudocode for token-level steering provided in Algorithm 1. Hyperparameters for training and experimental setup are documented in the Appendix A. Experiments are conducted with the Gemma-2-2B-IT model and pre-trained Gemma Scope SAE (16K features). Results are averaged over multiple validation samples, and robustness is assessed through layer-wise coefficient analysis (detailed layer-wise figures in Appendix A), feature diversity analysis (Appendix A.11), and task-wise comparisons of CRL-Token and CRL-Layer (Appendix A.6). All resources will be released publicly upon acceptance. These resources enable independent reproduction and extension of our work.

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

# A APPENDIX

## A.1 KEY TERMINOLOGY AND IMPLEMENTATION DETAILS

- **CRL-Token**: MDP formulated over token sequence at a single layer. State is residual stream activation $\mathbf{x}_t$ at token position $t$ for a fixed layer $\ell$. Uses separate policy/critic networks per layer.
- **CRL-Layer**: MDP formulated over layers at a single token position. State is layer-specific residual stream $\mathbf{x}^{(\ell)}$ at layer $\ell$ for a fixed token. Uses shared policy/critic networks across all layers.
- **Corrected samples**: Samples where CRL steering changes incorrect predictions to correct ones
- **Misguided samples**: Samples where CRL steering changes correct predictions to incorrect ones
- **Policy network** $\pi_\theta$: 2-layer MLP (Tanh activation) that maps residual stream activations to SAE feature selection probabilities (hidden dimension matching LLM)
- **Critic network** $V_\phi$: 2-layer MLP (Tanh activation) that estimates state value function for PPO training (hidden dimension matching LLM)
- **Steering coefficient** $c$: Scalar multiplier controlling intervention magnitude in $\tilde{x}_t = x_t + c \cdot a_t W_{dec}$
- **Constrained decoding**: Generation restricted to valid answer tokens (e.g., A/B/C/D for MMLU)
- **Unconstrained decoding**: Generation allowing any token output, including invalid responses

## A.2 ALGORITHM 1: CRL TRAINING PROCEDURE

---
**Algorithm 1** CRL Training Procedure

---
**Require:** Base model $M$, SAE ($W_{enc}$, $W_{dec}$), task dataset $D$
**Ensure:** Trained policy $\pi$, value function $V$
1: Initialize policy $\pi(a|s)$ and value function $V(s)$
2: **for** each episode **do**
3:     Sample batch $B$ from dataset $D$
4:     Initialize mask for AFM
5:     **for** each token position $t$ **do**
6:         Extract residual activation $x_t$
7:         Policy computes logits $\boldsymbol{\mu} = \pi_\theta(x_t)$
8:         Select feature $a_t = \text{TopK}(\boldsymbol{\mu} \odot \text{mask}^{(t)}, k)$
9:         Apply intervention: $\tilde{x}_t = x_t + c \cdot a_t \cdot W_{dec}$
10:        Compute natural SAE activations $z_t = \text{SAE}(x_t)$
11:       Update mask: $\text{mask}_i^{(t+1)} = \text{mask}_i^{(t)} \vee [\mathbf{z}_{t,i} > 0]$
12:       Compute reward $r_t$ from task outcome
13:     **end for**
14:     Update $\pi$ and $V$ using PPO with rewards $\{r_t\}$
15: **end for**
16: **return** trained $\pi$, $V$

---

## A.3 TASK-SPECIFIC REWARDS

We use the following reward structures for each task:

**Benchmarks with ground-truth answers:**

- **MMLU:** Binary reward: +1 for correct multiple-choice answer, 0 for incorrect
- **MMLU-Pro:** Binary reward: +1 for correct multiple-choice answer, 0 for incorrect
- **GSM8K:** Binary reward: +1 for exact numerical match, 0 otherwise
- **BBQ Ambig:** Ternary reward: +1 for unbiased answer, -1 for biased answer, 0 for ambiguous
- **BBQ Disambig:** Ternary reward: +1 for unbiased answer, -1 for biased answer, 0 for ambiguous

**Benchmarks requiring evaluation models:**

- **HarmBench:** Binary reward: +1 for refusing harmful request, 0 for compliance. The DistilRoBERTa rejection classifier[2] identifies refusal vs compliance.

- **XSTest:** Binary reward: +1 for refusing harmful request, 0 for compliance. The Distil-RoBERTa rejection classifier identifies refusal vs compliance.

- **SimpleQA:** Binary reward: +1 if semantic similarity $\geq 0.6$, 0 otherwise. A ModernBERT STS model[3] matches generated answers against expected responses.

## A.4 EXPERIMENTAL SETUP

**Models and SAEs.** We conduct experiments using Gemma-2 2B-IT (Team, 2024a) with pre-trained SAEs from Gemma Scope (Lieberum et al., 2024) (16K features) across layers 1-26, and LLaMA-3.1 8B (Team, 2024b) with LLaMA Scope SAEs (He et al., 2024) (32K features). Both SAE families employ JumpReLU activation (Rajamanoharan et al., 2024) and were the only releases providing SAEs across all transformer layers at the time of writing. SAEs are transferable across fine-tuned models with low reconstruction loss. **Training Protocol.** Training uses PPO with batch size 8, evaluating on 500 validation samples every 100 training steps. Selection tasks use 1 token generation, while HarmBench/XSTest use 32 tokens and GSM8K uses 1024 tokens. For datasets with fewer than 4000 samples, we repeat training data to reach 4000 samples.

**Evaluation Benchmarks.** We evaluate CRL across five categories: **Knowledge**: MMLU (Hendrycks et al., 2021), MMLU-Pro (Wang et al., 2024). **Reasoning**: GSM8K (Cobbe et al., 2021). **Bias**: BBQ (Parrish et al., 2022). **Factuality**: SimpleQA (Wei et al., 2024). **Safety**: HarmBench (Mazeika et al., 2024), XSTest (Röttger et al., 2024).

## A.5 NON-RL BASELINES

To isolate the benefit of learned policy selection, we evaluated three non-RL baselines with 3 random seeds each:

- **Random Feature Selection**: Uniform random selection from the full SAE dictionary
- **Random Feature with AFM**: Random selection constrained to the Adaptive Feature Masking
- **Top SAE Feature by Activation**: Greedy selection of the highest naturally activated feature

Table 2: Non-RL baseline comparison across five methods on single-token (BBQ Ambig) and multi-token (HarmBench) generation tasks. Results averaged over 3 seeds.

| Method | Non-steered | Random | Random+AFM | Top Feature | CRL |
|---|---|---|---|---|---|
| BBQ Ambig | $60.17_{\pm0.01}$ | $58.36_{\pm0.03}$ | $60.16_{\pm0.01}$ | $59.94_{\pm0.25}$ | $\mathbf{65.86_{\pm3.03}}$ |
| HarmBench | $41.46_{\pm9.05}$ | $45.35_{\pm0.35}$ | $46.96_{\pm0.17}$ | $48.03_{\pm2.32}$ | $\mathbf{49.12_{\pm1.59}}$ |

CRL outperforms baselines on both tasks, demonstrating the learned policy does more than selecting highly activated features. Random Feature can degrade performance (-1.81 on BBQ Ambig), while Random+AFM outperforms Random Feature, confirming AFM's effectiveness without policy learning.

## A.6 CRL-LAYER: LAYER-WISE STEERING MECHANISM

For layer-wise steering, we extend the framework to operate across multiple transformer layers simultaneously using a single token position. The shared Markov Decision Process operates on layer-specific variables where all components are parameterized with layer index $\ell$.

Let $\mathbf{x}^{(\ell)} \in \mathbb{R}^d$ denote the residual stream activation at layer $\ell$ for a single target token position. Given layer-specific SAE components with encoder $\mathbf{W}_{enc}^{(\ell)} \in \mathbb{R}^{d \times d_{dict}}$ and decoder $\mathbf{W}_{dec}^{(\ell)} \in \mathbb{R}^{d_{dict} \times d}$, the steering mechanism applies perturbations across all layers:

---

[2]https://huggingface.co/ProtectAI/distilroberta-base-rejection-v1
[3]https://huggingface.co/dleemiller/ModernCE-base-sts

$$\tilde{\mathbf{x}}^{(\ell)} = \mathbf{x}^{(\ell)} + \mathbf{a}^{(\ell)}\mathbf{W}_{dec}^{(\ell)} \tag{10}$$

where $\mathbf{a}^{(\ell)} \in \{0,1\}^{d_{dict}}$ is the layer-specific action vector. The shared MDP coordinates decisions across layers through a joint policy:

$$\pi_\theta(\mathbf{a}^{(1:L)}|\mathbf{x}^{(1:L)}) = \prod_{\ell=1}^{L} \pi_\theta^{(\ell)}(\mathbf{a}^{(\ell)}|\mathbf{x}^{(\ell)}) \tag{11}$$

where $L$ is the total number of layers and each layer-specific policy $\pi_\theta^{(\ell)}$ shares parameters while adapting to layer-specific representations.

**CRL-Layer Results.**

CRL-Layer's practical advantage lies in sharing policy and critic networks across layers. This approach reduces computational resources compared to extending CRL-Token to multiple layers without cross-layer compatibility. Although representation spaces for each layer's residual stream differ, residual connections enforce shared vector spaces, and existing layer-wise reuse approaches (Ye et al., 2021; Elhoushi et al., 2024; Raposo et al., 2024) support network sharing across layers.

Table 3: Performance comparison between CRL variants across different benchmarks on Gemma 2 2B model. Results show accuracy (%).

| Method | MMLU | MMLU-Pro | BBQ Ambig | BBQ Disambig | XSTest | HarmBench |
|---|---|---|---|---|---|---|
| Baseline | 52.23 | 30.30 | 59.10 | 75.42 | 86.35 | 44.64 |
| CRL-Layer | 55.00 | 29.18 | 62.93 | 75.68 | 86.98 | 71.43 |
| CRL-Token | 55.55 | 30.44 | 61.88 | 76.73 | 86.98 | 50.25 |

Table 3 presents a comparison between CRL-Layer, and CRL-Token across multiple benchmarks. CRL-Layer achieves improvements over baseline across most tasks except MMLU-Pro, though it generally underperforms CRL-Token on complex tasks, with the exception of refusal tasks. This pattern reveals that while layer-wise sharing provides clear computational advantages, it introduces limitations for complex tasks requiring layer-specific weights. Notably, CRL-Layer achieves the highest performance on HarmBench, outperforming both CRL-Token, suggesting that layer-shared refusal patterns benefit from cross-layer feature consistency.

## A.7 LLaMA Results

We extend our evaluation to LLaMA-3.1 8B model with LLaMA Scope SAEs to demonstrate the generalizability of our approach across different model architectures.

Table 4: Performance results for LLaMA-3.1 8B model across different tasks using CRL-Token approach. MMLU and MMLU-Pro use 0-shot evaluation, BBQ tasks use 1-shot evaluation, SimpleQA uses 0-shot evaluation, XSTest and HarmBench use refusal rate evaluation.

| Task | Type | Layer | Before | After | Improvement |
|---|---|---|---|---|---|
| MMLU | knowledge | 20 | 61.41 | 62.33 | +0.92 |
| MMLU-Pro | knowledge | 26 | 32.13 | 33.16 | +1.03 |
| BBQ Ambig | bias | 7 | 83.97 | 85.04 | +1.07 |
| BBQ Disambig | bias | 30 | 90.07 | 90.16 | +0.09 |
| SimpleQA | factuality | 25 | 0.43 | 0.96 | +0.53 |
| XSTest | safety | 23 | 61.27 | 59.37 | -1.90 |
| HarmBench | safety | 5 | 0.71 | 6.07 | +5.36 |

Table 4 demonstrates that CRL-Token generalizes across different model families, with mixed results across tasks. While most tasks show improvements (knowledge, bias, factuality), XSTest exhibits a decline (-1.90), suggesting task-specific sensitivity to the steering approach. HarmBench shows improvement (+5.36), consistent with the Gemma-2 2B results, demonstrating safety steering across

architectures. The mixed results confirm that our approach transfers across different model architectures while highlighting the importance of task-specific optimization, maintaining the interpretable feature-based control mechanism.

## A.8 CRITIC VALUE ANALYSIS

**Multi-token Generation Task Analysis.**

The four sample categories (correct/incorrect, corrected/misguided) are analyzed for HarmBench and XSTest tasks to understand critic behavior patterns in multi-token generation scenarios.

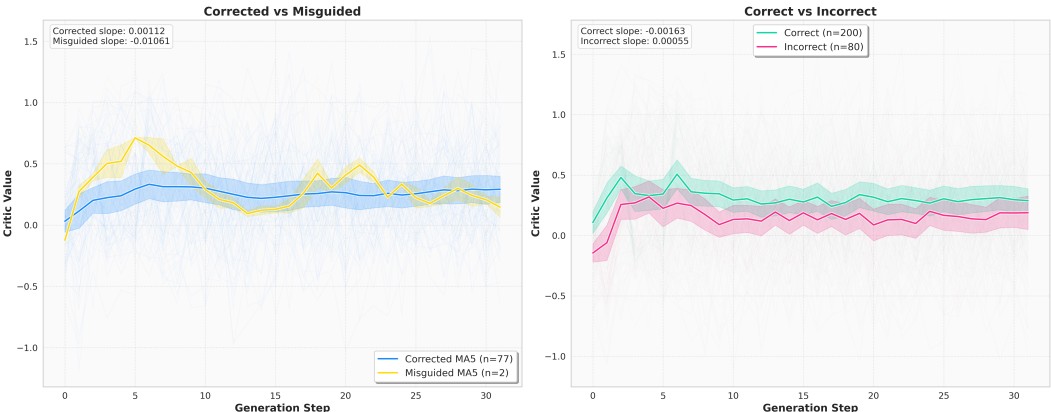

Figure 10: Critic network learning curve for HarmBench task, demonstrating convergence patterns.

Our analysis focuses on linear regression gradient slopes and the visible gaps between sample categories. Empirically, we observed that critic value gaps between correct and incorrect samples become more pronounced in later layers. We observe two distinct patterns in critic trajectories analogous to those in single-token generation tasks. These patterns reflect either errors in critic value estimation (manifesting as bias gaps) or limitations in the policy's critic value utility (affecting gradient dynamics).

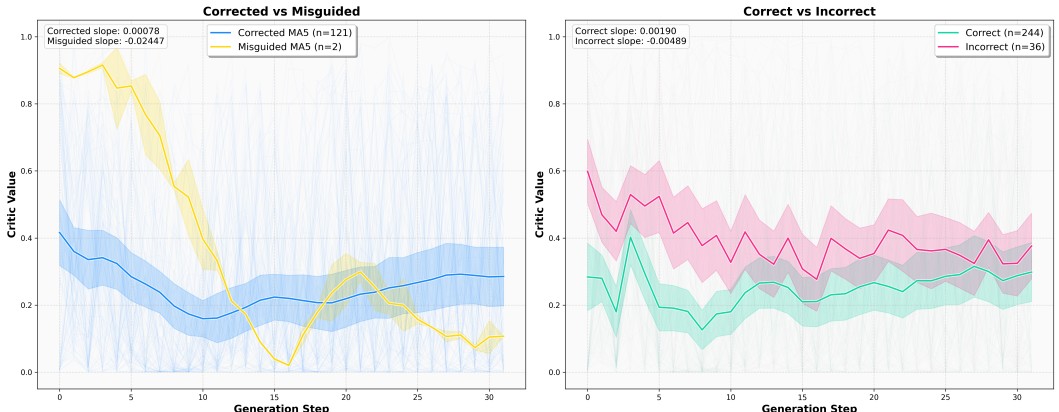

Figure 11: Critic network learning curve for XSTest task, demonstrating convergence patterns.

For HarmBench (Figure 10), the critic successfully distinguishes between correct and incorrect samples, whereas XSTest (Figure 11) exhibits inverted estimation patterns. However, while HarmBench shows clear gaps at sequence boundaries with minimal gradient differences, XSTest demonstrates estimation errors in gap measurements but maintains clear superiority for correct samples in slope analysis across both correct/incorrect and corrected/misguided categories.

## A.9 HALLUCINATION MITIGATION ANALYSIS

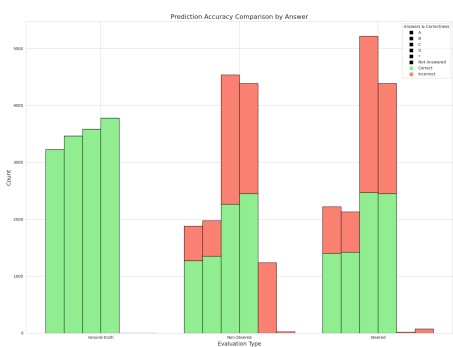

Figure 12: MMLU hallucination answers comparison between baseline and CRL-Token steering, demonstrating effective elimination of invalid responses outside the provided options (A, B, C, D).

For single-token generation tasks, particularly MMLU without constrained decoding, a substantial portion of performance improvement stems from hallucination mitigation. CRL-Token steering effectively eliminates responses that fall outside the provided answer options (A, B, C, D).

The baseline model frequently generates invalid responses such as "*" or whitespace instead of selecting from the given options. This behavior significantly impacts performance metrics, as these responses are automatically marked incorrect regardless of the underlying reasoning quality. Our CRL-Token approach addresses this issue by learning to constrain outputs to valid answer choices while maintaining the model's reasoning capabilities.

The analysis reveals that a portion of performance gains in MMLU tasks can be attributed to this hallucination mitigation effect, with the remainder coming from improved feature selection and reasoning enhancement. This finding highlights the importance of output format consistency in evaluation benchmarks and demonstrates CRL's effectiveness in learning task-specific constraints.

## A.10 FEATURE SELECTION EVALUATION

We analyze the policy network's feature selection patterns to understand which SAE features contribute most to performance improvements. For each feature $i$, we compute:

### A.10.1 IMPACT SCORE

$$n_i = \sum_t \mathbf{1}\{i \in S_t\}, \quad N = \sum_j n_j, \tag{12}$$

$$p_i = \frac{n_i}{N}, \tag{13}$$

$$c_i = \sum_t \mathbf{1}\{i \in S_t\}\,\mathbf{1}\{\text{improved}_t\}, \quad m_i = \sum_t \mathbf{1}\{i \in S_t\}\,\mathbf{1}\{\text{degraded}_t\}, \tag{14}$$

$$\text{Impact}_i = \frac{(c_i + m_i) \cdot \log(n_i + \varepsilon)}{n_i \cdot \log(n_i + \varepsilon)} = \frac{c_i + m_i}{n_i}, \quad (\varepsilon > 0) \tag{15}$$

where $S_t$ is the set of selected features at step $t$ (e.g., top-$k$), and $\text{improved}_t$/$\text{degraded}_t$ denote whether steering improved/degraded the outcome versus baseline.

### A.10.2 FEATURE DIVERSITY

Additionally, we compute feature diversity to understand the policy's exploration behavior:

$$\text{Feature Diversity} = -\sum_{i=1}^{d_{dict}} p_i \log p_i \tag{16}$$

where $p_i$ is the probability of selecting feature $i$. Higher entropy indicates more diverse feature usage across the SAE dictionary. Together, these metrics quantify how diverse the trained policy's feature selection is and how much the selected features actually influence behavior change.

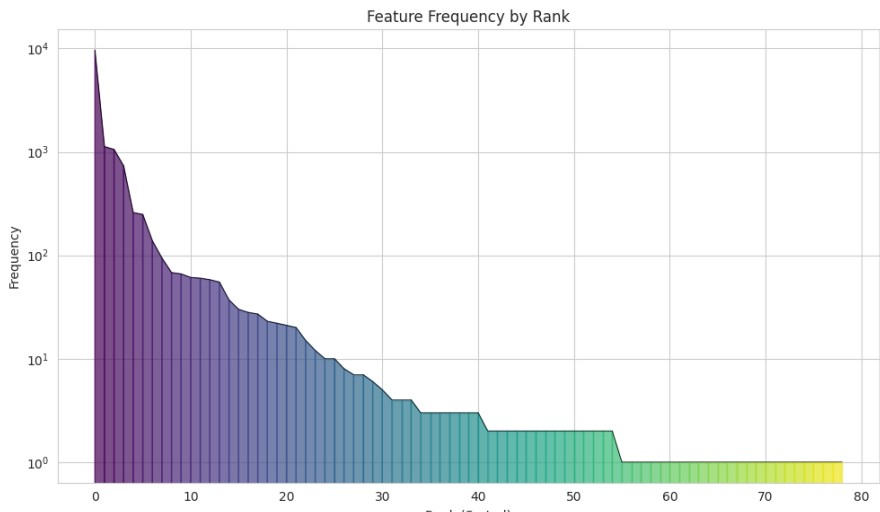

Figure 14: Feature usage hierarchy in Gemma 2 2B MMLU task, demonstrating that the policy selectively identifies specific features rather than random selection.

### A.11 FEATURE DIVERSITY AND IMPACT SCORE

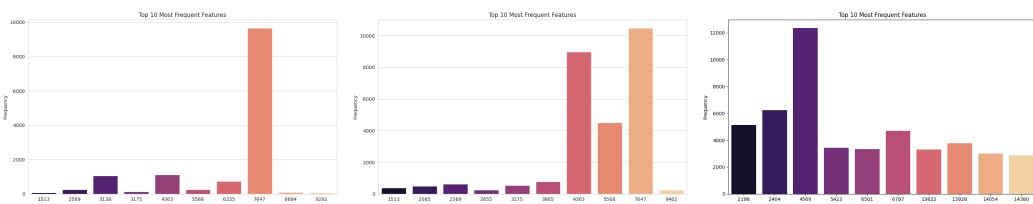

Figure 13: Top activated features for MMLU (left) and BBQ Disambig (center) and GSM8K (right) tasks, revealing semantic patterns in feature selection for reasoning and bias mitigation.

Feature diversity remains consistent across different hyperparameters within the same task, though it decreases with increased policy layer depth. As shown in Figure 13, tasks requiring longer token generation horizons (GSM8K: 6.695) or higher complexity (MMLU-Pro: 5.476) demonstrate elevated feature diversity compared to shorter response tasks (HarmBench: 2.938), with XSTest (4.757) showing intermediate diversity levels. Impact scores exhibit an inverse relationship with feature diversity: tasks with lower diversity show higher concentrated impact scores (HarmBench: 0.808), while high-diversity tasks demonstrate more distributed feature contributions (XSTest: 0.357, MMLU-Pro: 0.625). This pattern suggests that complex reasoning tasks require broader feature engagement, whereas focused tasks benefit from concentrated feature activation. Figure 14 demonstrates that activation frequency exhibits rank-dependent decay with task-specific distribution patterns reflecting underlying diversity characteristics.

### A.12 LAYER-WISE ANALYSIS

Figure 15 shows residual stream norm growth across layers for Gemma 2 2B model on the MMLU task. This pattern is consistent with prior observations that residual stream norms increase with network depth (Heimersheim & Turner, 2023).

The analysis for layers $\ell \geq 18$ for BBQ tasks (Figure 18) confirms that ambiguous and disambiguated contexts require different optimal intervention layers. This suggests that bias mitigation strategies must be adapted to the specific type of ambiguity present in the input. The layer-wise analysis shows similar patterns across tasks: later layers generally provide more effective intervention points, while

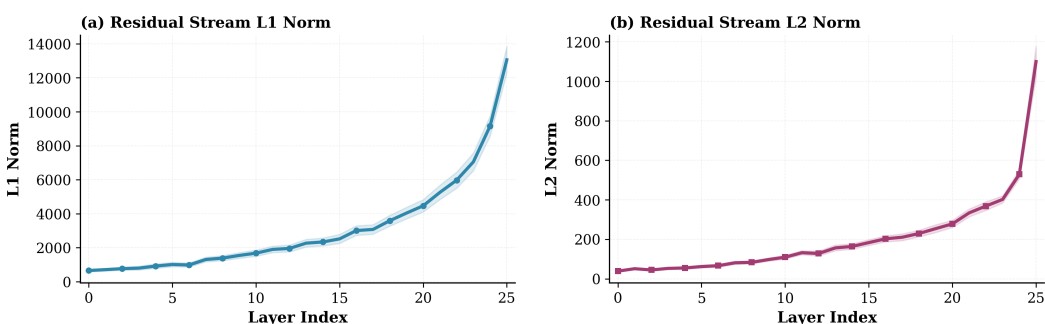

Figure 15: Norm growth along layers for Gemma 2 2B model in MMLU task.

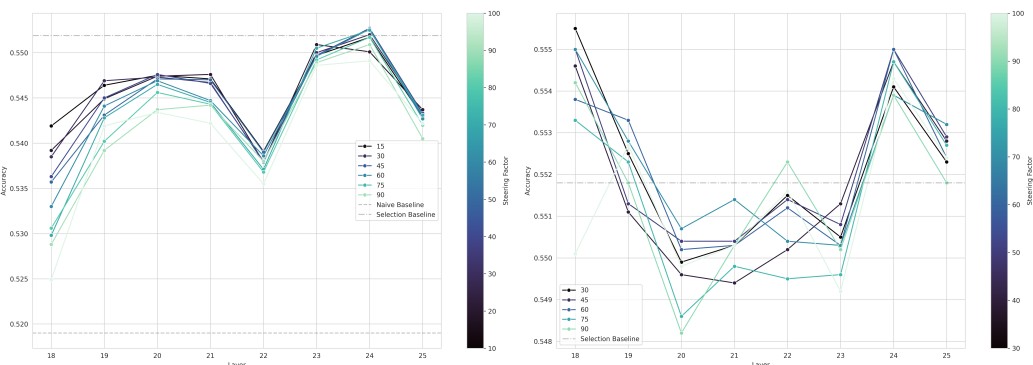

Figure 16: Extended MMLU layer analysis for layers $\ell \geq 18$, showing unconstrained (left) and constrained (right) decoding patterns.

early layers show degraded performance with large coefficients. This pattern holds across different coefficient values and task types, consistent with observations that residual stream norms increase across layers (Heimersheim & Turner, 2023).

## B   FEATURE ANALYSIS DETAILS

### B.1   CORRECTED FEATURES FOR GSM8K TASK

The following features demonstrate positive steering effects, correcting model outputs from incorrect to correct responses:

- **4504** diagrams and schematic representations related to workflows, communication systems, and protocols (correct/incorrect: 321/170, corrected/misguided: 15/4)
- **406** specific mentions of individuals, organizations, and their respective roles or activities (correct/incorrect: 116/113, corrected/misguided: 12/4)
- **1440** instances of the word "vice" and related terms (correct/incorrect: 161/119, corrected/misguided: 11/4)
- **14204** indicators of group dynamics and power relations (correct/incorrect: 248/179, corrected/misguided: 16/6)
- **10961** terms related to statistical methods and implementation details (correct/incorrect: 223/182, corrected/misguided: 20/8)
- **2699** significant concepts related to problems and solutions within a defined framework (correct/incorrect: 234/187, corrected/misguided: 17/7)
- **2317** phrases indicating comparisons or relationships between two entities or elements (correct/incorrect: 120/86, corrected/misguided: 12/5)
- **2720** mathematical expressions and equations related to functions and inequalities (correct/incorrect: 151/150, corrected/misguided: 11/5)

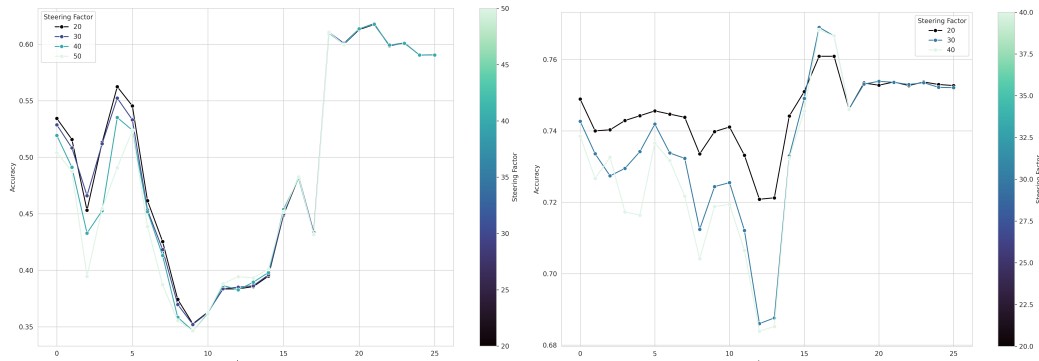

Figure 17: BBQ layer analysis for ambiguous (left) and disambiguated (right) tasks across different coefficient values.

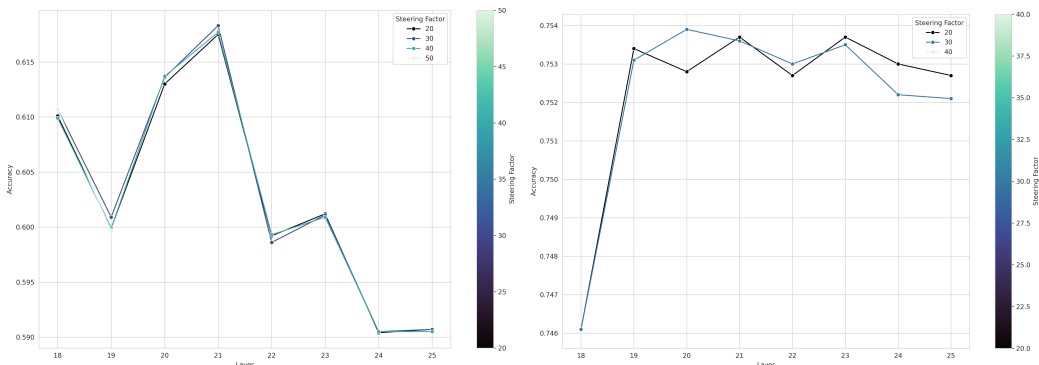

Figure 18: BBQ layer analysis for layers $\ell \geq 18$ for ambiguous (left) and disambiguated (right) tasks.

- 347 phrases and concepts related to accountability and compliance (correct/incorrect: 148/129, corrected/misguided: 15/7)
- 7708 mathematical operations and expressions in various forms (correct/incorrect: 188/217, corrected/misguided: 19/9)

## B.2 ADDITIONAL CORRECTED EXAMPLE

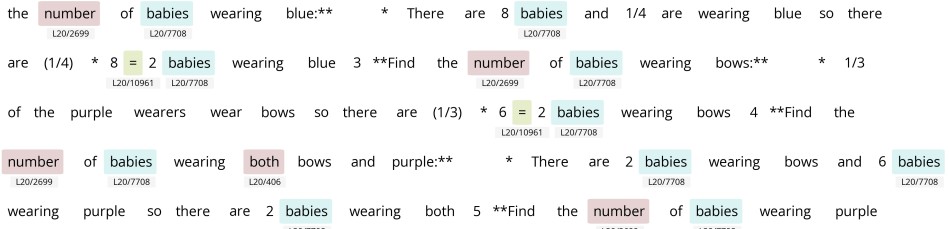

Figure 19: Additional GSM8K corrected example: Feature 7708, described as *mathematical operations and expressions in various forms*, activates on quantitative units like "babies" and "packs", demonstrating lexical generalization beyond surface forms.

## B.3 MISGUIDED FEATURES FOR GSM8K TASK

Conversely, these features demonstrate negative steering effects, inadvertently changing correct responses to incorrect ones:

- **7999** mathematical expressions or equations (correct/incorrect: 40/42, corrected/misguided: 1/10)
- **4069** mathematical notation and geometric properties related to circles and angles (correct/incorrect: 121/100, corrected/misguided: 7/16)
- **13752** elements related to job creation and economic context (correct/incorrect: 109/78, corrected/misguided: 5/10)
- **10968** specific coding or mathematical terms and phrases related to programming or data analysis (correct/incorrect: 99/112, corrected/misguided: 8/16)
- **16295** significant terms and phrases related to scientific studies, particularly in the context of law, medicine, and research methodologies (correct/incorrect: 99/93, corrected/misguided: 6/11)
- **12174** references to understanding and problem-solving processes (correct/incorrect: 180/157, corrected/misguided: 10/17)
- **15434** complex relationships involving socio-legal and psychological themes (correct/incorrect: 113/125, corrected/misguided: 8/13)
- **1222** sentences expressing emotional vulnerability and complex interpersonal dynamics (correct/incorrect: 74/61, corrected/misguided: 7/11)
- **7834** indicative symbols and formatting used in programming or mathematical expressions (correct/incorrect: 97/101, corrected/misguided: 9/14)
- **3415** patterns of conditional phrases and expressions of uncertainty (correct/incorrect: 92/108, corrected/misguided: 8/12)

### B.4 ADDITIONAL MISGUIDED EXAMPLE

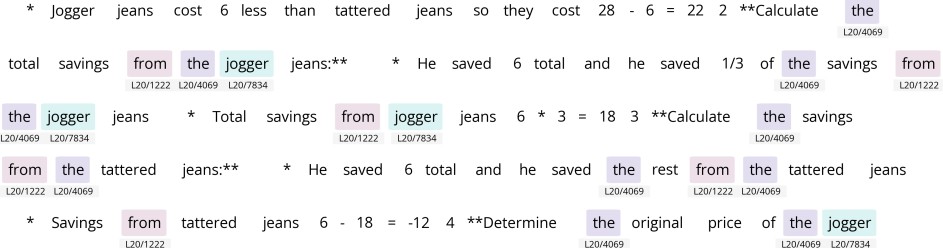

Figure 20: GSM8K misguided example 3: Feature interference causes abandonment of correct solution path.

### B.5 ADDITIONAL BRANCH TRACKING ANALYSIS

We analyze decision bifurcations at critical tokens, comparing layer 10 and layer 20 feature selections across GSM8K tasks. The following cases show different steering outcomes depending on which layer's features are selected.

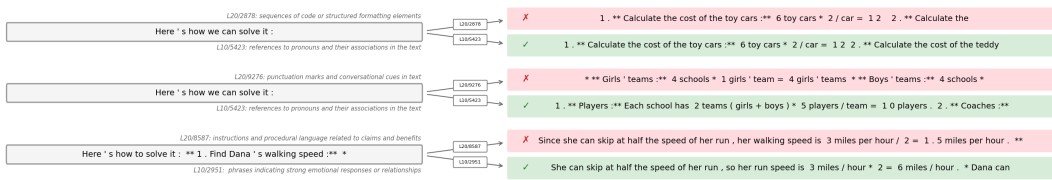

Figure 21: Cases where layer 10 features succeed. Top: Toy calculation where L10's "pronoun references" captures concrete items while L20's "code formatting" misapplies structure. Middle: Tournament problem where L10's "pronoun references" handles entity tracking while L20's "punctuation cues" misidentifies structure. Bottom: Speed calculation where L10's "emotional relationships" correctly solves the problem while L20's "claims and benefits" misapplies procedural framing.

In these three cases, L10 features produce correct outputs while L20 features fail. Both the toy calculation and tournament problems use L10's "pronoun references" feature, which generates accurate enumerated calculations, whereas L20's "code formatting" and "punctuation cues" features

produce incorrect results. The speed conversion case shows L10's "emotional relationships" feature generating the correct formula ("3 miles/hour * 2 = 6"), while L20's "claims and benefits" produces wrong logic ("3 miles per hour / 2 = 1.5"). Feature labels alone do not predict which layer succeeds: the "emotional relationships" label bears no obvious relation to speed calculation.

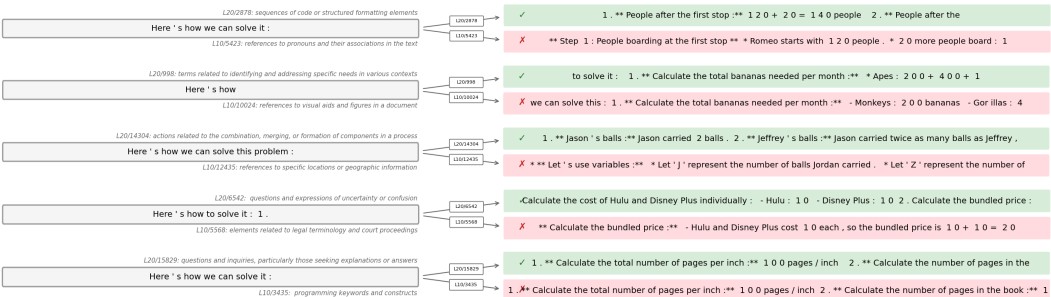

Figure 22: Cases where layer 20 features succeed. Top: Bus problem where L20's "code formatting" correctly structures sequential steps while L10's "pronoun references" fails. Second: Banana calculation where L20's "needs identification" captures requirement semantics while L10's "visual aids" produces incorrect output. Third: Variable abstraction where L20's "combination and formation" proceeds with direct calculation while L10's "geographic information" chooses variable-based approach. Fourth: Subscription pricing where L20's "uncertainty expressions" captures choice scenarios while L10's "legal proceedings" produces incorrect output. Bottom: Pages calculation where L20's "questions and inquiries" captures exploratory reasoning while L10's "programming constructs" treats it as code.

In these five cases, L20 features produce correct outputs while L10 features fail. The bus passenger problem shows L20's "code formatting" generating correct sequential tracking ("1. **People after the first stop:** 120 + 20 = 140"), while L10's "pronoun references" produces verbose, incorrect narrative. The variable abstraction case shows L10 choosing a variable-based approach ("Let 'J' represent...") while L20 proceeds directly with numerical calculation ("Jason carried 2 balls"). In the subscription pricing problem, L10's "legal contracts" generates incorrect output, while L20's "uncertainty expressions" handles the choice scenario. The banana and pages problems similarly show L10 features producing incorrect outputs, whereas L20 features generate correct multi-step solutions.

**Observations.** Across these eight examples, we observe a pattern: L10 succeeds on three cases involving discrete item counting and direct arithmetic, while L20 succeeds on five cases involving multi-step accumulation, choice scenarios, and variable reasoning. L10 failures show patterns of surface matching (e.g., financial terminology leading to "legal contracts", notation leading to "programming constructs"), whereas L20 failures show structural misapplication to simple problems (e.g., "code formatting" on toy calculation). Feature labels provide limited predictive value—"emotional relationships" solves speed calculations, "code formatting" performs both correctly and incorrectly on different tasks. These divergent outcomes demonstrate that layer choice affects steering effectiveness, which CRL's layer-specific policy exploits to adapt to different problem structures.

