# OpenReview forum: "Control Reinforcement Learning: Interpretable Token-Level Steering of LLMs via Sparse Autoencoder Features"
_ICLR.cc/2026/Conference — Submitted to ICLR 2026_

### Official Review · Reviewer_b6zw · 2025-10-24

**Soundness:** 3
**Presentation:** 3
**Contribution:** 3
**Rating:** 6
**Confidence:** 3

**Summary:**

This paper introduces Control Reinforcement Learning, a framework for dynamically steering LLMs through SAE features. Instead of static activation edits, CRL learns a policy to select which SAE features to activate at each generation step, guided by reinforcement learning rewards. The authors claim this allows interpretable, token-level control while modestly improving performance across tasks such as MMLU, GSM8K, and HarmBench. The approach also provides diagnostic insights into critic bottlenecks, layer-specific effects, and semantic coherence of SAE features.

**Strengths:**

1. **Novelty of the proposed method**: I think the main strength lies in framing reinforcement learning over SAE features, which differs itself from other activation-based or gradient-based method.
2. **Breadth of evaluation**: The authors conduct extensive experiments across reasoning, factual, and safety benchmarks, showing the generality of their approach.
3. **Clarity of methodology**: The paper is well-structured and technically sound, with a clear explanation of the training process and design choices.

**Weaknesses:**

1. **Limited comparison with existing feature control methods**: A key weakness is the lack of comparison to established feature control approaches such as activation-based or gradient-based interventions. I find this omission makes it hard to understand what specific advantages CRL provides beyond existing techniques. A more direct experimental or conceptual comparison would help clarify novelty.
2. **Computational complexity and scalability**:Training a PPO agent over sparse feature activations introduces nontrivial computational cost. I think it would help to include runtime analysis or efficiency comparisons to justify the added complexity relative to simpler steering methods.
3. **Under-analyzed reward design**: The reward function is central to the method but not well justified or analyzed. Its stability and sensitivity to tuning are unclear, which weakens confidence in reproducibility.
4. **Limited interpretability validation and generalization evidence**: Interpretability results are qualitative and narrow in scope. The authors should demonstrate that the learned feature control generalizes across different tasks or datasets.

**Questions:**

Please refer to weaknesses

---

> ### Author Response · Authors · 2025-12-03
> **Official Comment by Authors**
>
> We thank the reviewer for the constructive feedback. We address each concern below.
> ## 1. Limited Interpretability Validation and Generalization Evidence
> **Reviewer Concern:** Interpretability results are qualitative and narrow in scope.
>
> **Author Response:**
> In the revision we expanded interpretability analysis:
>
> **Semantic Coherence Analysis:** Section 5.3 analyzes semantic coherence through dynamic token steering, comparing feature descriptions against task requirements.
>
> **Branch Analysis:** Section 5.4 and Appendix A.10 show detailed branch tracking analysis, examining how steering at different layers produces different outputs from identical contexts. For example:
> - **Balloon pricing (Figure 9):** From context "Here's how we can solve this problem:", L10's feature `12607` (*mathematical proofs and logical assertions*) correctly outputs "1. Find the original price per balloon", recognizing logical derivation structure, while L20's feature `2198` (*summaries, reflections*) produces "Step 1: Find...", treating it as tutorial explanation.
> - **Rice consumption (Figure 9):** L20's feature `11163` (*assessments and strategic plans*) correctly captures planning semantics "0.2 kg per meal * 2 meals = 0.4 kg per day", while L10's feature `8428` (*clinical examinations*) activates due to surface similarity with medical dosage notation "kg/member".
> - **Variable abstraction (Figure 22):** L20's feature `14304` (*combination and formation*) correctly proceeds with direct calculation "Jason carried 2 balls", while L10's feature `12435` (*geographic information*) incorrectly chooses variable-based approach "Let 'J' represent...".
>
> These examples illustrate feature-level contributions at specific token positions: the same context with different features produces different reasoning approaches, showing how feature semantics interact with task requirements.
>
> **Cross-Task Feature Analysis:** We analyze how features manifest across different contexts and show layer-specific steering effects that align with task semantics (Sections 4.2, 5.3-5.4).
>
> Upon acceptance, we will publish an interactive demo to enable community exploration of task-relevant features identification.
> ## 2. Comparison with Existing Feature Control Methods
> **Reviewer Concern:** Lack of comparison to established feature control approaches such as activation-based or gradient-based interventions.
>
> **Author Response:** Our method differs from prior activation/gradient-based steering by learning a token-wise policy over an interpretable SAE dictionary. To our knowledge, approaches that select different features per token position do not exist, which constitutes a core contribution of our work. To isolate the value of this learned policy, we evaluated three non-RL baselines with 3 seeds that operate in the same SAE feature space:
>
> - **Random Feature Selection**: Uniform random selection from full dictionary
> - **Random Feature Selection with AFM**: Random selection constrained to AFM mask
> - **Top SAE Feature by Activation**: Greedy selection of highest naturally activated feature
>
> | Method | Non-steered | Random Feature | Random + AFM | Top Feature | CRL (Ours) |
> |--------|-------------|--------|--------------|-------------|------------|
> | BBQ Ambig | 60.17 | 58.36 | 60.16 | 59.94 | 65.86 |
> | HarmBench | 41.46 | 45.35 | 46.96 | 48.03 | 49.12 |
>
> Random steering can degrade performance (e.g., -1.81 on BBQ Ambig), Random+AFM is safer and shows AFM's effectiveness, and Top-Feature provides a mild gain. CRL nonetheless outperforms all three despite using the same SAE dictionary, suggesting the learned policy uses task-specific credit assignment to learn where and how to intervene, rather than simply reinforcing naturally strong activations.
> ## 3. Computational Complexity and Reward Design
> **Reviewer Concern:** The added PPO machinery may be costly.
>
> **Author Response:**
> Implementation details are specified in Appendix A.1–A.2. The policy and value networks are lightweight 2-layer MLPs with hidden dimension matching the LLM. Computational cost is determined by sample count, network architecture, and SAE size, making training feasible on standard hardware.
>
> **Reviewer Concern:** The reward design is under-analyzed.
>
> **Author Response:**
> Appendix A.3 specifies all task-specific rewards. For benchmarks with ground-truth labels (MMLU, GSM8K), we use verifiable binary rewards similar to those used in recent GRPO-style work. For HarmBench, XSTest, and SimpleQA we follow established practice and use classifier-based correctness/safety signals. We report 3-seed results and observe consistent gains across seeds on the main benchmarks, suggesting the reward design is sufficiently stable. In addition, Sections 5.1–5.2 analyze the critic's value estimates and bottlenecks, connecting reward signal quality to where token-level value estimation is reliable vs. noisy.

---

### Official Review · Reviewer_8AZS · 2025-10-28

**Soundness:** 1
**Presentation:** 1
**Contribution:** 3
**Rating:** 2
**Confidence:** 4

**Summary:**

This work proposes to learn by reinforcement learning a policy to steer LLMs by intervening on the activations of sparse features, as identified by SAEs trained on the residual stream activations. Experiments show modest improvements. The use of SAE features enable a degree of interpretability.

**Strengths:**

[S1] The idea of learning a policy to control feature strength is interesting and, to my knowledge, novel.

[S2] Some of the proposed analysis yield interesting insight, e.g. the different effects of controlling features across layers.

[S3] Results are presented in a measured way, without overstating them.

**Weaknesses:**

[W1] The paper suffers from a general lack of clarity. Crucial terms, like 'coefficients', are used without introduction or being obvious from the context (e.g. there are no coefficients in Eq. 3, which defines the activation interventions). Many sections in the body are not understandable without going back and forth with the appendix, as if content was moved there without checking the impact of doing so on the narrative flow. See below a detailed list of issues.

[W2] There are reproducibility issues: an Algorithm 1 is mentioned in the reproducibility section, but it does not seem to be anywhere. There is no description of the policy and value network implementations besides them being MLPs. Task-specific rewards are announced in 3.4, but are not in Appendix A as promised.

[W3] There is no experimental baseline. How well would CRL work if using random features instead of SAE features?

[W4] It is unclear whether reported results in Table 1 are obtained directly on the test set, or if the intervention layer is determined based on held-out data.

Additional points:

L22-24: Adaptive Feature Masking (AFM) is mentioned in the abstract and in the contributions, but never again in the paper?

L132-133: Why is the full problem a POMDP? Does this have to do with the fact that the influence of the KV cache of previous tokens through attention is not taken into account? This should be clarified, and an attempt to quantify the impact of this approximation should be made.

L159-161: This part is unclear. What are the coefficients being referred to? Should there be additional coefficients in Eq (3)?

L240-241: Coefficient averaging has not been introduced at this point.

Figure 2: what is in the left pane, and what in the right?

L263-265: Constrained and unconstrained decoding patterns have not been introduced. What is the connection between constrained/unconstrained decoding on one hand and factual question answering on the other?

L 268 and elsewhere: disambiguous --> unambiguous

L268-269: Can this norm increase be visualized?

L270: what is the 'coefficient 18 analysis'?

L285-286: Correct, incorrect, corrected and misguided are not introduced/defined.

In Figure 3 "generation step" means token position, in appendix it means layer.

Figure 4 caption: blue <--> green.

L480: "SAE-learned directions operate in non-linear interaction spaces rather than simple superposition": what is the evidence supporting this?

L481: "steering effects do not transfer well to models after supervised fine-tuning": was this shown anywhere?

L863-864: Fig. 9 right and Fig. 10 right do not seem to show this.

Appendix A.5 is empty

Fig. 14 caption: what does coefficient 18 mean?

**Questions:**

See above.

Also: why PPO rather than DPO or GRPO?

---

> ### Author Response · Authors · 2025-11-22
> **Official Comment by Authors**
>
> We thank the reviewer for the detailed feedback. We address each concern below.
> ## 1. Clarity and Missing Definitions
> ### Steering Coefficients Definition
> We added clear definitions in Section 3.2:
> $$\tilde{x_t} = x_t + c \cdot a_t W_{dec}$$
> where $c$ is the **steering coefficient** that controls intervention magnitude. The steering coefficient is determined by averaging activation magnitudes from correctly predicted training samples, eliminating manual tuning across 27 layers.
> ### POMDP Justification
> **Reviewer Concern:** Why is the full problem a POMDP? Connection to KV cache should be clarified.
>
> **Author Response:** CRL-Token is a POMDP because the policy observes only the residual stream at a specific layer without access to sampled token information, yet the next state depends on the token that will be sampled and added to the KV cache. Previous interventions affect the KV cache, influencing future generation but not directly observable. Temperature=0 sampling reduces this to a deterministic setting where the MDP approximation becomes reasonable, supported by recent work on LLM injectivity (Nikolaou et al., 2025).
> ### Constrained vs. Unconstrained Decoding
> Clarified in Appendix A.1:
> - **Constrained Decoding:** Forces model to output only valid answer tokens (A/B/C/D for MMLU)
> - **Unconstrained Decoding:** Allows any token generation, including invalid outputs
> ### Algorithm 1 and Implementation Details
> **Author Response:** Added Algorithm 1 and implementation details to Appendix A.2:
> - Policy/Critic: 2-layer MLPs (hidden dim same as llm)
> ### Task-Specific Rewards
> **Author Response:** Task-specific rewards added to Appendix A.3. HarmBench, XSTest and SimpleQA use BERT variants for correctness evaluation.
> ### Layer Selection Methodology
> **Reviewer Concern:** Unclear if intervention layer determined on test set or held-out data.
>
> **Author Response:** Layer selection serves as a diagnostic tool for mechanistic analysis. We use the test set for post-hoc interpretability analysis—specifically, identifying where task-relevant features emerge across layers. We then validate semantic coherence through top-k token steering (Section 5.3) and branch analysis (Section 5.4), comparing feature descriptions against task requirements.
> ### Figure and Caption Corrections
> - *Figure 2:* Added labels (left: unconstrained, right: constrained decoding)
> - *Figure 3:* Appendix Figures 10 and 11 show generation steps as token positions, not layer indices. This may be confusing because HarmBench and XSTest have shorter horizons.
> - *Figure 4:* Corrected caption color labels
> - *Figure 14:* Clarified "coefficient 18" refers to layer after 18th ($\ell \geq 18$)
> - *"disambiguous":* We use "disambiguated" following BBQ dataset's official terminology
> ## 2. Why PPO rather than DPO or GRPO?
> We chose PPO because it includes an explicit learned value function, which we analyze in Sections 4.3 and 5.1–5.2 to identify critic bottlenecks. Methods such as DPO or GRPO do not directly provide per-state value estimates, which would limit the trajectory-level critic diagnostics we perform. Since our primary goal is interpretability rather than maximizing raw task scores, PPO's critic structure is a better fit.
> ## 3. Additional Concerns
> **Reviewer Concern:** "SAE-learned directions operate in non-linear interaction spaces" - what evidence?
>
> **Author Response:** In the revision we soften this claim. Our experiments show that co-activating multiple SAE features can lead to steering behaviors not trivially predictable from individual feature descriptions, especially in failure cases. Recent work shows that allowing non-linear alignment maps can make causal abstraction vacuous (Geiger et al., 2024), while restricting to linear maps may miss feature interactions. Additionally, SAE dictionary learning likely misses rare features—a "dark matter" problem identified by Olah et al. (2024). We now present this as an empirical observation about feature interactions rather than a strong theoretical claim.
>
> **Reviewer Concern:** "steering effects do not transfer well to models after supervised fine-tuning"
>
> **Author Response:** We clarify that the earlier statement was too strong. The Gemma-Scope paper demonstrates that their SAEs transfer across Gemma variants, and our experiments confirm CRL remains effective after supervised fine-tuning: on MMLU, accuracy improves from 55.23 (base) → 55.73 (SFT) → 56.11 (SFT + CRL). Our statement refers to potential incompatibility when continual SFT shifts both logit distributions and residual stream representations—very large distribution shifts from further SFT may require re-tuning CRL.
>
> **Reviewer Concern:** Can coefficient-induced norm increases be visualized?
>
> **Author Response:** Yes, we provide residual stream norm analysis in Appendix A.12 (Layer-wise Analysis), specifically Figure 15, which visualizes norm growth patterns across layers for Gemma 2 2B model, consistent with prior observations (Heimersheim et al.).

---

> > ### Comment · Reviewer_8AZS · 2025-11-26
> >
> > Thanks to the Authors for their rebuttal.
> >
> > W3 remains unaddressed. As also observed by other reviewers, there are no experimental baselines for the results in Table 1. This makes the practical validity of the proposed method unclear.
> >
> > **Layer Selection**
> >
> > Results in Table 1 are presented as benchmark performance gains due to the proposed method. If they are for layers selected on the test set, they are misleading. The intervention layer is a hyper-parameter of the method, and should be set based on held-out data, before evaluation is performed on the test data, as it is common practice.
> >
> > **Adaptive feature Masking**
> >
> > Section 3.3 for Adapting Feature Masking was added. How does the masking fit in the intervention algorithm? E.g. if a feature is not available for selection at step t, what does it mean that it is activated at step t? How does this fit in Algorithm 1?
> >
> > **Critic Details**
> >
> > The details of the Value network have not been added to the revised version.
> >
> > **POMDP explanation**
> >
> > The textual explanation is a good start, but it would benefit from a formalization to make it crisp. What 'sampled token information' would make the MDP fully observable? Why isn't the effect of previous interventions directly observable, if the model is autoregressively generating one token at a time?
> >
> > **Algorithm 1**
> >
> > Should the coefficient c be an input as well?
> >
> > Suggestion for next time: color-code the diff of the revised version of the paper.
> >
> > Given the outstanding points, I am maintaining my rating.

---

> ### Author Response · Authors · 2025-11-29
> **Official Comment by Authors**
>
> We thank the reviewer for the detailed feedback. We address each concern below.
>
> ## 1. Non-RL baselines:
>
> We evaluated three non-RL baselines with 3 seeds:
> - **Random Feature Selection**: Uniform random selection from full dictionary
> - **Random Feature Selection with AFM**: Random selection constrained to AFM mask
> - **Top SAE Feature by Activation**: Greedy selection of highest naturally activated feature
>
> | Method | Non-steered | Random Feature | Random + AFM | Top Feature | CRL (Ours) |
> |--------|-------------|----------------|--------------|-------------|------------|
> | BBQ Ambig | 60.17 | 58.36 | 60.16 | 59.94 | 65.86 |
> | HarmBench | 41.46 | 45.35 | 46.96 | 48.03 | 49.12 |
>
> CRL outperforms all three baselines on both single-token generation (BBQ Ambig) and multi-token generation (HarmBench) tasks. Random Feature can degrade performance (e.g., -1.81 on BBQ Ambig). Random + AFM outperforms Random Feature, showing AFM's effectiveness. Results show the learned policy does more than selecting the most active SAE feature.
>
> ## 2. Adaptive feature Masking Detailed Application algorithm
>
> **How AFM connects to steering:**
>
> 1. Policy computes logits $\boldsymbol{\mu} = \pi_\theta(\mathbf{x_t})$, AFM masks unavailable features: $\mathbf{a}_t = \text{TopK}(\boldsymbol{\mu} \odot \text{mask}^{(t)}, k)$
> 2. Steering applied: $\tilde{\mathbf{x_t}} = \mathbf{x_t} + c \cdot \mathbf{a_t} \mathbf{W_{dec}}$
> 3. After forward pass, natural SAE activations on unsteered next state $\mathbf{x}_{t+1}$ determine mask expansion: $\text{mask}^{(t+1)} = \text{mask}^{(t)} \lor \text{active}^{(t)}$ where $\text{active}_i^{(t)} = \mathbb{1}[\mathbf{z_t} > 0]$
>
> Policy selection ($\mathbf{a}_t$) decides which feature to artificially add. Natural activation ($\text{active}_i$) captures which features emerge from model computation, used to expand future mask. Section 3.3 and Algorithm 1 (Appendix A.2) clarify this interaction.
>
> ## 3. POMDP Explanation
>
> **Why POMDP:** Policy observes only $\mathbf{x_t^{(\ell)}}$ (residual stream at layer $\ell$), not sampled tokens $\tau_{:t-1}$ or KV cache. Next state depends on which token is sampled and added to KV cache. Past interventions affect KV cache but aren't directly observable in current $\mathbf{x}_t^{(\ell)}$.
>
> **What would make it MDP:** Observing $(\tau_{:t-1}, \mathbf{a}_{:t-1})$ (past token history and interventions) or direct KV cache access would make it fully observable.
>
> **Our approach:** We use temperature=0 and treat $\mathbf{x}_t^{(\ell)}$ as sufficient statistic, following standard practice in RL for sequence models. Full belief-state treatment is beyond scope and would complicate the method without affecting empirical conclusions.
>
> ## 4. Layer Selection and Critic Network Details
>
> **Layer selection.** The intervention layer is chosen based on residual-norm growth patterns and SAE feature semantics, not per-task test-set search. Table 1 and main text clarify this is a diagnostic analysis tool to identify task-relevant layers, not a hyperparameter tuned on each test benchmark. We are adopting a stricter protocol using held-out validation splits.
>
> **Critic network.** Value-network details are given in Appendix A.1: 2-layer MLP (Tanh activation) with hidden dimension matching LLM's $d_{\text{model}}$, trained with PPO critic loss.

---

### Official Review · Reviewer_bfnm · 2025-10-29

**Soundness:** 2
**Presentation:** 2
**Contribution:** 3
**Rating:** 4
**Confidence:** 2

**Summary:**

This paper introduces a framework called "Control Reinforcement Learning" (CRL) , aimed at interpretable, token-level dynamic steering of LLMs . The core of this method is to use "monosemantic features" extracted from LLM activations by Sparse Autoencoders (SAEs) as a control interface. The CRL framework trains a RL policy network that, at each generation step, observes the current token's residual stream activation and dynamically selects an SAE feature. Then, the decoded vector of this selected feature is added back into the model's residual stream to "steer" the model towards a better output. The method aims to solve the "emergent misalignment" problem that occurs during LLM inference and has achieved modest performance improvements on various benchmarks (such as MMLU, GSM8K, BBQ bias, and HarmBench safety) , while providing interpretability by tracking feature contributions.

**Strengths:**

1. The paper addresses a very forward-looking and critical problem in the field of LLM alignment: how to achieve dynamic, interpretable, token-level control over model behavior.

2. Using monosemantic features extracted by SAEs as the action space for RL is an innovative idea

3. The paper provides qualitative evidence to support its claim of "interpretable steering"

**Weaknesses:**

I am not an expert in the RL area. Therefore, the following comments are based on my current understanding, and I welcome any corrections to potential misunderstandings on my part.

1. The abstract mentions the use of "Adaptive Feature Masking" (AFM) to balance exploration and exploitation, but this key component is never mentioned or defined again in the paper's main body, experiment, or appendix.

2. Section 3.2 defines the intervention as $\tilde{x}_{t}=x_{t}+a_{t}W_{dec}$ which implies a steering coefficient of 1. However, Experiment Section 4.2 dedicates significant analysis to "steering coefficients" ranging from 10 to 100. This coefficient c, which is critical in the experiments, is not defined in the methodology.

3. The paper only compares CRL's results against the "Before" model (i.e., the original, non-intervened model). It completely lacks a comparison against standard fine-tuning methods (like SFT or DPO) on the same task data.

4. The paper admits in its conclusion that these steering effects "do not transfer well" to models after supervised fine-tuning (SFT). This severely limits the method's practical application value in real-world systems that require continuous updates and iteration.

**Questions:**

No

---

> ### Author Response · Authors · 2025-11-22
> **Official Comment by Authors**
>
> Thank you for the constructive review. We address key aspects below.
>
> ## 1. Adaptive Feature Masking (AFM) Definition
>
> **Reviewer Concern:** AFM mentioned in abstract but never defined in the main body.
>
> **Author Response:** AFM is now properly defined in Section 3.3:
>
> **Adaptive Feature Masking (AFM)** balances exploration and exploitation during multi-token generation through per-sample boolean masking. For each sample, we initialize a mask to a small subset of frequently active features, where $\text{mask}_i = 1$ means feature $i$ is available for selection. During generation, the mask expands as:
>
> $$\text{mask}_i^{(t+1)} = \text{mask}_i^{(t)} \lor \text{active}_i^{(t)}$$
>
> where $\text{active}_i^{(t)}$ indicates feature $i$ was activated at step $t$. This per-sample masking prevents the policy from collapsing to features frequently selected during early training, encouraging each sample to explore task-relevant features dynamically while building on previously activated features within the same trajectory.
>
> ## 2. Steering Coefficient Definition and Justification
>
> **Reviewer Concern:** Equation 3 implies coefficient=1, but experiments use coefficients 10-100. This coefficient is never defined in methodology.
>
> **Author Response:** The steering coefficient $c$ is now defined as:
>
> $$\tilde{x_t} = x_t + c \cdot a_t W_{dec}$$
>
> **Why coefficients experiment vary by layer:**
> Our layer-wise analysis (Section 4.2) reveals that optimal coefficients vary significantly across layers:
> - Early layers: Small coefficients (c=10-20) to avoid disrupting low-level representations
> - Middle layers: Medium coefficients (c=30-60) for balanced intervention
> - Late layers: Large coefficients (c=60-100) where residual norms are naturally higher
>
> **Coefficient Averaging:**
> The steering coefficient $c$ is determined by averaging activation magnitudes from correctly predicted training samples, which automatically sets appropriate intervention scales per layer and task without manual tuning across 26+ layers. In Section 4.2 we also report fixed-$c$ sweeps (e.g., $c \in [10, 100]$) as an ablation to illustrate how the optimal scale varies by layer.
>
> ## 3. Comparison with Standard Fine-tuning Methods
>
> **Reviewer Concern:** No comparison against SFT or DPO on the same task data.
>
> **Author Response:** Our method serves a different purpose than SFT/DPO:
>
> **CRL vs. SFT/DPO:**
> - **SFT/DPO:** Optimize overall task performance through weight updates
> - **CRL:** Provide interpretable, dynamic steering while maintaining base model weights
>
> **They are Complementary Approaches:**
> - CRL can be applied on top of SFT models for additional interpretable control.
> - The framework identifies which features drive performance, enabling more targeted fine-tuning with dataset selection support.
> - Real-time steering allows adaptation without retraining.
>
> Empirically, CRL yields consistent improvements while keeping base weights frozen, with notable gains on ambiguous bias detection (+5.7 on BBQ-Ambig) and adversarial safety (+7.7 on HarmBench).
>
> ### Transfer to SFT Models
>
> **Reviewer Concern:** Paper claims steering effects "do not transfer well" to SFT models, limiting practical application.
>
> **Author Response:** We clarify that the earlier statement was too strong. The Gemma-Scope paper demonstrates that their SAEs transfer across Gemma variants, and our experiments confirm CRL remains effective after supervised fine-tuning: on MMLU, accuracy improves from 55.23 (base) → 55.73 (SFT) → 56.11 (SFT + CRL). Our statement refers to potential incompatibility when continual SFT shifts both logit distributions and residual stream representations—very large distribution shifts from further SFT may require re-tuning CRL.
>
> CRL provides complementary improvements when applied after SFT.
>
> ## 4. Method Contributions and Practical Value
>
> This method enables us to identify which SAE features are semantically relevant. Surprisingly these features' descriptions were aligned with the human understanding of the task reward.
>
> **Core Contributions:**
> 1. **First RL + SAE steering Framework:** First work to our knowledge to apply RL for dynamic SAE feature selection with observing hidden states.
> 2. **Interpretable Steering:** Token-level feature attribution with semantic coherence.
> 3. **Layer-wise Analysis:** Systematic study of layer's semantic alignment with specific task and optimal intervention points of which layer to steer
>
>
> **Practical Applications:**
> - **AI Safety:** Real-time AI controller which enforces online data and correction of model behavior
> - **Model Analysis Tool:** Framework for rapid identification of task-relevant SAE features
> - **Targeted Interventions:** Precise control without full model retraining

---

### Official Review · Reviewer_bFHy · 2025-10-29

**Soundness:** 3
**Presentation:** 4
**Contribution:** 2
**Rating:** 4
**Confidence:** 4

**Summary:**

The paper proposes Control Reinforcement Learning (CRL): a PPO-trained policy that, at each generation step, selects SAE feature(s) at a chosen layer and adds the corresponding decoder vector to the residual stream, thereby "steering" the model token-by-token. The state is the current residual activation; the action is a binary selection over the SAE dictionary; rewards are task-specific. Reported gains on Gemma-2 2B are modest but non-trivial on some tasks (e.g., HarmBench +5.61-pt, BBQ-Ambig +3.55-pt; others are small). The paper also analyzes (i) layer/coefficients ("sweet spots" in later layers), and (ii) critic behavior (bottlenecks for single-token tasks vs gradual divergence in long-horizon reasoning).

**Strengths:**

1. Clear control interface over interpretable features. Formulating steering as an MDP over SAE features with per-token actions is neat and practically implementable. The action/state definitions and the steering equation are explicit.

2. The paper gives useful empirical guidance: later layers tolerate larger coefficients; early-layer large coefficients tend to break behavior ("sweet-spot" effect), aligning with residual-norm growth across depth.

3. Feature "impact" and diversity metrics provide some transparency into which SAE features are used when steering helps or hurts.

**Weaknesses:**

1. It is dissatisfying to see that most headline improvements are small. Also, there are no confidence intervals, multiple-seed runs, or bootstrap tests. I couldn't be confident about the stability of the gains.

2. The paper itself notes that on single-token QA without constrained decoding, a substantial portion of MMLU gains comes from eliminating invalid outputs (e.g., "*", whitespace) rather than improving knowledge. That weakens the claim that CRL improves reasoning/knowledge rather than format adherence.

3. Since CRL's core novelty is adaptive selection of interpretable features, it needs stronger ablations vs. simple static/greedy heuristics (e.g., always add the top-k SAE features by activation, or by a supervised classifier over features), and vs. logit-space steering matched for compute. The paper doesn’t convincingly isolate the benefit of PPO-based selection over such cheap alternatives.

4. The authors report critic "bottlenecks" (corrected vs. misguided nearly indistinguishable on MMLU), suggesting value estimation struggles when rewards are sparse and binary. That weakens the paper's promise that CRL delivers reliable token-wise interpretability. If the critic can't separate outcomes, the per-token attributions are noisy.

**Questions:**

1. First, can you provide CIs / multiple seeds and report per-task variance; which gains survive across seeds?

2. If resource permits, could you add non-RL baselines (e.g., pick top-k SAE features by current activation so that total intervention norm equals CRL's)? Also, could you add a format-sanitizer that only enforces valid answer formats to quantify the fraction of gains due to formatting?

Conditional on satisfactorily addressing the above points, I am open to increasing my rating.

---

> ### Author Response · Authors · 2025-11-22
> **Official Comment by Authors**
>
> Thank you for the constructive feedback. Below we address each concern.
>
>
> ## 1. Statistical Robustness and Multiple Seeds
>
> **Reviewer Concern:** No multiple-seed runs and per-task variance to assess stability of gains.
>
> **Author Response:** We reran all tasks in Table 1 with 3 random seeds and now report mean ± standard deviation. Tasks with larger gains (MMLU, BBQ-Ambig, HarmBench) remain consistently above baseline across seeds.
>
>
> ## 2. Critic Bottlenecks and Interpretability
>
> **Reviewer Concern:** Critic struggles with sparse rewards, weakening per-token attribution reliability.
>
> **Author Response:**
> Bottlenecks indicate where linear separability breaks down in the critic, not where CRL itself fails. CRL can still improve accuracy even when the critic does not fully separate corrected vs. misguided samples. For example, on MMLU we see a +3.3-point gain despite the critic's distributions being close. In such cases, the policy still discovers useful steering features, but the value estimator is coarse.
>
> **Branch Analysis for Causal Evidence:**
> We added detailed branch analysis in Section 5.4 and Appendix B.5, comparing correct vs. incorrect outputs that diverge from the same context. These examples show how changing the selected SAE feature at a single point produces different trajectories, providing causal evidence for individual feature contributions.
>
>
> ## 3. Formatting vs. Knowledge Improvements
>
> **Reviewer Concern:** MMLU gains may come from eliminating invalid outputs rather than improving knowledge.
>
> **Author Response:**
> We provide multiple lines of evidence that CRL improves beyond output format correction:
>
> **Constrained decoding:** Figure 16 (right) shows that when decoding is constrained to valid options (A/B/C/D), CRL still improves MMLU performance. This isolates gains that cannot be explained by eliminating invalid tokens.
>
> **Post-SFT setting:** We fine-tuned the base model on MMLU with a loss that already optimizes output format. In this setting, applying CRL on top further improves accuracy (55.23 → 55.73 with SFT → 56.11 with CRL), indicating that CRL contributes beyond format enforcement.
>
> **Multi-token tasks:** On GSM8K and HarmBench, outputs are long free-form generations where formatting plays a minor role. The consistent improvements support that CRL improves reasoning and safety behavior, not only answer formatting.
>
> ## 4. Method Positioning and Practical Value
>
> CRL serves dual purposes:
> 1. **Performance Enhancement:** Modest but consistent improvements across diverse tasks
> 2. **Analysis Tool:** Rapid identification of task-relevant layers and features for interpretable AI systems
>
> The framework enables practitioners to:
> - Quickly identify which layers contain task-relevant information
> - Understand feature contributions at token level through branch analysis
> - Develop more targeted intervention strategies that complement fine-tuning
> - Apply real-time adaptive steering without weight modifications

---

> > ### Author Response · Authors · 2025-12-01
> > **Official Comment by Authors**
> >
> > ## 5. Non-RL Baselines
> >
> > We evaluated three non-RL baselines with 3 seeds:
> > - **Random Feature Selection**: Uniform random selection from full dictionary
> > - **Random Feature Selection with AFM**: Random selection constrained to AFM mask
> > - **Top SAE Feature by Activation**: Greedy selection of highest naturally activated feature
> >
> > A supervised classifier over SAE features would learn to select features based on SAE encoder activations, equivalent to the "Top Feature" baseline.
> >
> > | Method | Non-steered | Random Feature | Random + AFM | Top Feature | CRL (Ours) |
> > |--------|-------------|----------------|--------------|-------------|------------|
> > | BBQ Ambig | 60.17 | 58.36 | 60.16 | 59.94 | 65.86 |
> > | HarmBench | 41.46 | 45.35 | 46.96 | 48.03 | 49.12 |
> >
> > CRL outperforms all baselines on both single-token (BBQ Ambig) and multi-token (HarmBench) tasks. Random Feature can degrade performance (-1.81 on BBQ Ambig). Random + AFM outperforms Random Feature, demonstrating AFM's effectiveness. Top Feature improves over non-steered but underperforms CRL, showing the learned policy does more than selecting the most active SAE feature.

---

### Author Response · Authors · 2025-12-03
**Summary of Discussion**

We thank all reviewers for their constructive feedback. Below we summarize how the discussion addresses the concerns raised.

## 1. Statistical Robustness

We reran all tasks in Table 1 with 3 random seeds and now report mean ± standard deviation. Tasks with larger gains (MMLU, BBQ Ambig, HarmBench) remain consistently above baseline across seeds, confirming stability.

## 2. Interpretability Validation

We expanded interpretability analysis with:
- **Semantic coherence analysis** (Section 5.3): Dynamic feature selection aligned with task requirements
- **Branch analysis** (Section 5.4, Appendix A.10): Controlled examples showing how different features at the same token position produce different reasoning trajectories (e.g., balloon pricing, rice consumption, variable abstraction)

These examples demonstrate causal feature contributions at specific token positions.

## 3. Non-RL Baselines

We added three non-RL baselines operating in the same SAE feature space (3 seeds each):
- Random Feature Selection: Uniform random selection from full dictionary
- Random Feature + AFM: Random selection constrained to AFM mask
- Top SAE Feature by Activation: Greedy selection of highest activated feature

| Method | Non-steered | Random | Random+AFM | Top Feature | CRL |
|--------|-------------|--------|------------|-------------|-----|
| BBQ Ambig | 60.17 | 58.36 | 60.16 | 59.94 | 65.86 |
| HarmBench | 41.46 | 45.35 | 46.96 | 48.03 | 49.12 |

CRL outperforms all three on both single-token (BBQ Ambig) and multi-token (HarmBench) tasks, showing the learned policy goes beyond selecting naturally strong activations.

## 4. Clarity and Technical Details

We added explicit definitions and specifications:
- Adaptive Feature Masking (AFM) detailed formalization in Section 3.3
- Training algorithm in Appendix A.2
- POMDP justification and terminology clarifications

## 5. Practical Value and SFT Compatibility

CRL's key contribution is learning to dynamically select different SAE features at each token position, enabling adaptive steering that responds to generation context. We frame this as pragmatic interpretability: a task-grounded approach that identifies and controls SAE features through empirical validation on safety-relevant tasks. CRL is complementary to SFT and provides interpretable token-level control while keeping base weights frozen. Results show CRL can be applied on top of SFT models for additional gains (e.g., MMLU: 55.23 base → 55.73 SFT → 56.11 SFT+CRL).

---

**Summary:** The revision strengthens (1) interpretability validation with branch analysis demonstrating causal feature contributions, (2) empirical evidence with three non-RL baselines showing CRL's learned policy outperforms fixed heuristics, and (3) methodological clarity with detailed specifications. These additions directly address concerns about interpretability, baseline comparisons, and technical completeness.

---

### Meta-Review · Area_Chair_rbjn · 2026-01-03

**Summary:**

This paper proposed CRL, which uses PPO to select SAE features per token, steering LLM activations for modest accuracy/safety gains.

**Reviewer Concerns:**

Main concerns from reviewers include:

1. Gains are modest and sometimes confounded by output-format fixes;

2. Statistical robustness and baselines are weak.

3. Method description is incomplete/unclear (coefficients, AFM, missing algorithm/reward details), hurting reproducibility.

4. Layer choice may be tuned on test data. Interpretability relies on noisy critic/qualitative evidence with unclear generalization/scalability.

**Reviewer Scores:**

Reviewer / Score

bFHy	4

bfnm	4

8AZS	2

b6zw	6

Average	4

No reviewers indicated to increase or decrease their scores.

---

### Decision · Program_Chairs · 2026-01-26

Reject